# scRNA-Seq reveals distinct stem cell populations that drive hair cell regeneration after loss of Fgf and Notch signaling

Mark E Lush[1†], Daniel C Diaz[1†], Nina Koenecke[1], Sungmin Baek[1], Helena Boldt[1], Madeleine K St Peter[1], Tatiana Gaitan-Escudero[1], Andres Romero-Carvajal[1,2], Elisabeth M Busch-Nentwich[3,4], Anoja G Perera[1], Kathryn E Hall[1], Allison Peak[1], Jeffrey S Haug[1], Tatjana Piotrowski[1]*

[1]Stowers Institute for Medical Research, Kansas City, United States; [2]Pontificia Universidad Catolica del Ecuador, Ciencias Biologicas, Quito, Ecuador; [3]Wellcome Sanger Institute, Wellcome Genome Campus, Hinxton, United Kingdom; [4]Department of Medicine, University of Cambridge, Cambridge, United Kingdom

**Abstract** Loss of sensory hair cells leads to deafness and balance deficiencies. In contrast to mammalian hair cells, zebrafish ear and lateral line hair cells regenerate from poorly characterized support cells. Equally ill-defined is the gene regulatory network underlying the progression of support cells to differentiated hair cells. scRNA-Seq of lateral line organs uncovered five different support cell types, including quiescent and activated stem cells. Ordering of support cells along a developmental trajectory identified self-renewing cells and genes required for hair cell differentiation. scRNA-Seq analyses of *fgf3* mutants, in which hair cell regeneration is increased, demonstrates that Fgf and Notch signaling inhibit proliferation of support cells in parallel by inhibiting Wnt signaling. Our scRNA-Seq analyses set the foundation for mechanistic studies of sensory organ regeneration and is crucial for identifying factors to trigger hair cell production in mammals. The data is searchable and publicly accessible via a web-based interface.
DOI: https://doi.org/10.7554/eLife.44431.001

*For correspondence:
pio@stowers.org

†These authors contributed equally to this work

Competing interests: The authors declare that no competing interests exist.

## Introduction

Non-mammalian vertebrates readily regenerate sensory hair cells during homeostasis and after injury, whereas in mammals hair cell loss leads to permanent hearing and vestibular loss (*Bermingham-McDonogh and Rubel, 2003*; *Brignull et al., 2009*; *Corwin and Cotanche, 1988*; *Cruz et al., 2015*; *Ryals and Rubel, 1988*). The molecular basis for the inability of mammals to trigger proliferation and a regenerative response is still unknown. Understanding hair cell production in regenerating species is essential for elucidating how regeneration is blocked in mammals. We and others showed that the zebrafish lateral line system is a powerful in vivo model to study the cellular and molecular basis of hair cell regeneration (*Kniss et al., 2016*; *Lush and Piotrowski, 2014b*; *Ma and Raible, 2009*; *Romero-Carvajal et al., 2015*; *Viader-Llargués et al., 2018*). The lateral line is a sensory system that allows aquatic vertebrates to orient themselves by detecting water motion. The lateral line organs (neuromasts), distributed on the head and along the body contain approximately 60 cells, composed of central sensory hair cells surrounded by support cells and an outer ring of mantle cells (*Figure 1A–B*). The lateral line system is one of the few sensory organs where stem cell behaviors can be observed at the single cell level in vivo because the organs are located in the skin of the animal, are experimentally accessible and easy to image. These properties make it an ideal system to

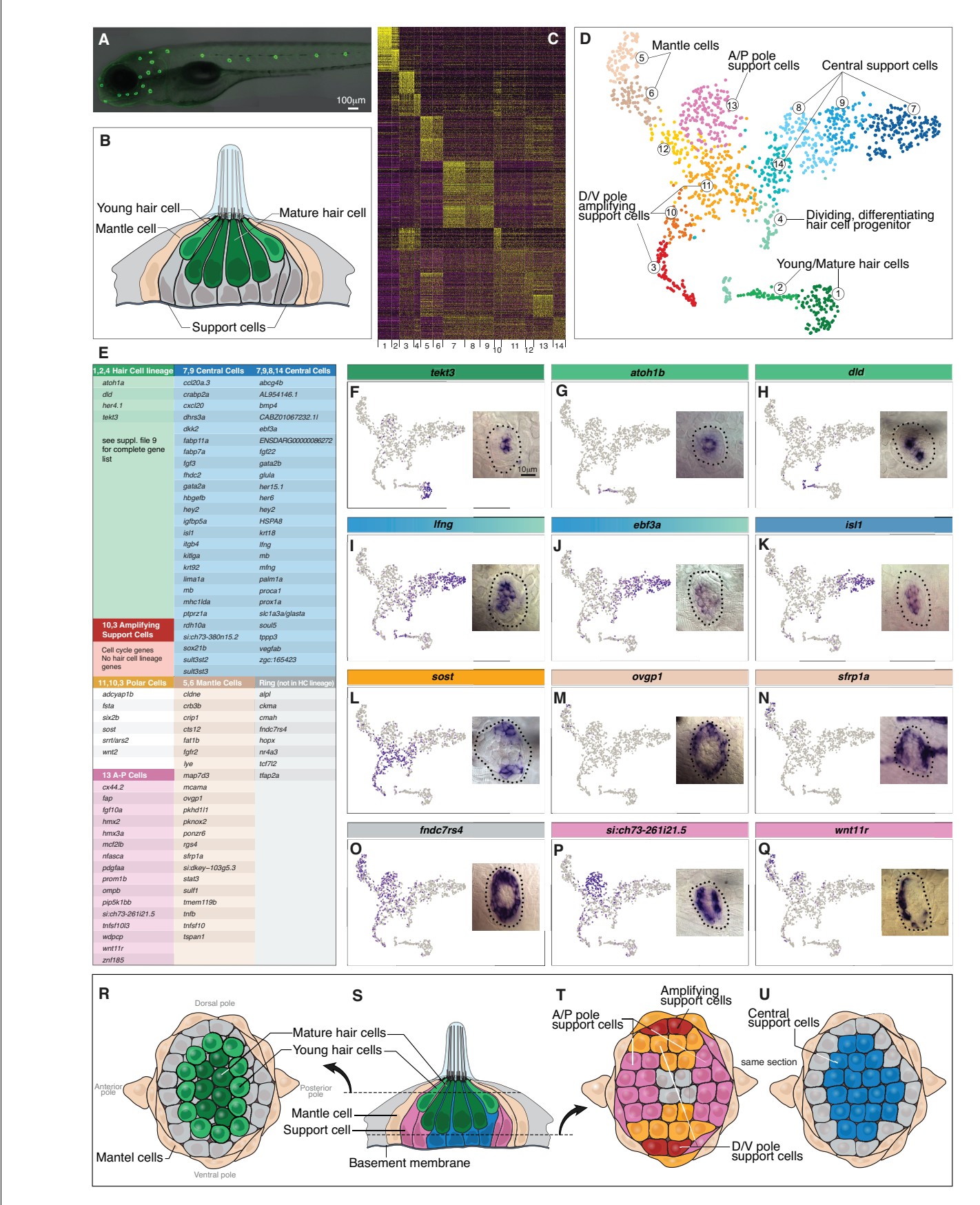

**Figure 1.** Single cell RNA-Seq reveals support cell heterogeneity. (**A**) *Et(Gw57a)* labels support cells with GFP. (**B**) Schematic of a cross section through a neuromast. (**C**) Heatmap showing the expression levels of the top 50 marker genes (y-axis) for each cluster (x-axis), sorted by highest fold change. (**D**) t-SNE plot showing the different cell clusters. (**E**) Table of marker genes that distinguish the different cell clusters. (**F–Q**) t-SNE plots of selected cluster markers and in situ hybridization with these genes. (**R, T and U**) Schematics of dorsal views of neuromasts with the different cell types colored. (**S**) Schematic of a cross section through the center of a neuromast.

DOI: https://doi.org/10.7554/eLife.44431.002

The following video is available for figure 1:

**Figure 1—video 1.** *Tg(prox1a:tagRFP;pou4f3:gfp)* during regeneration.

DOI: https://doi.org/10.7554/eLife.44431.003

study hair cell regeneration. Despite the unusual location of the hair cells on the trunk, lateral line and ear hair cells develop by similar developmental mechanisms. Importantly, the morphology and function of sensory hair cells are evolutionarily conserved from fish to mammals (*Duncan and Fritzsch, 2012*; *Nicolson, 2005*; *Whitfield, 2002*). For example, mutations in genes causing deafness in humans also disrupt hair cells in the zebrafish lateral line (*Nicolson, 2005*). We therefore hypothesize that the basic gene regulatory network required for hair cell regeneration could be very similar in zebrafish and mammals. In support of this hypothesis, our findings that Notch signaling needs to be downregulated to activate Wnt-induced proliferation during hair cell regeneration is also true in the mouse cochlea (*Li et al., 2015b*; *Romero-Carvajal et al., 2015*).

In zebrafish regeneration occurs via support cell proliferation and differentiation, and in chicken and amphibians hair cells regenerate from proliferating and transdifferentiating support cells (*Balak et al., 1990*; *Bermingham-McDonogh and Rubel, 2003*; *Harris et al., 2003*; *Jones and Corwin, 1996*; *Lush and Piotrowski, 2014b*; *Ma et al., 2008*; *Wibowo et al., 2011*; *Williams and Holder, 2000*). Yet, even in regenerating species, support cells are not well characterized due to a dearth of molecular markers and the lack of distinct cytological characteristics.

Using time-lapse analyses and tracking of all dividing cells in regenerating neuromasts, coupled with cell fate analyses, we previously identified two major support cell lineages: 1) support cells that divide symmetrically to form two progenitor cells (amplifying divisions); and 2) support cells that divide to form two hair cells (differentiating divisions) (*Romero-Carvajal et al., 2015*). A recent publication has confirmed these lineages (*Viader-Llargués et al., 2018*). The two cell behaviors display a striking spatial compartmentalization. Amplifying divisions occur in the dorsal-ventral (D/V) poles and differentiating divisions occur in the center. Mantle cells surrounding the support cells only divide after severe injury to the neuromasts but rarely divide if only hair cells are killed. In addition, we observed quiescent support cells. Thus, there are at least four support cell types in a neuromast that likely play different functions in balancing progenitor maintenance with differentiation to ensure the life-long ability to regenerate.

To identify the gene regulatory network that triggers support cell proliferation and hair cell differentiation, we previously performed bulk RNA-Seq of support cells at different time points during regeneration (*Jiang et al., 2014*). These studies revealed the dynamic changes in signaling pathway activations over time. However, the unexpected diversity and mosaicism in support cells that we discovered during fate analyses (*Romero-Carvajal et al., 2015*) was masked in the RNA-Seq analysis of pooled support cells. To determine how many support cell populations exist in a neuromast and to characterize their transcriptomes, we performed scRNA-Seq analysis on 1521 purified, homeostatic neuromast cells from a transgenic line. As the lateral line neuromasts are GFP-positive and only consist of about 60 cells we were able to purify a rich cohort of lateral line cells (25x coverage). Our analysis identified seven major neuromast cell populations, revealing genes that are specifically expressed in these cells and characterized the transcriptional dynamics of the process of differentiation and of progenitor maintenance. These results led to the hypothesis that some support cell populations are involved in signaling to trigger regeneration, which we tested by scRNA-Seq analyses of *fgf3* mutants that strikingly show increased proliferation and hair cell regeneration. Our scRNA-Seq analysis identified *fgf3* targets that we could not identify in bulk RNA-Seq analyses. Importantly, we show that Notch and Fgf signaling act in parallel and that both need to be downregulated together to induce efficient regeneration. Knowing the temporal dynamics and identity of genes required for proliferation and hair cell differentiation are essential for devising strategies to induce hair cell regeneration in mammals.

## Results

### Single cell RNA-Seq reveals support cell heterogeneity

We reasoned that transcriptional profiling of homeostatic neuromast cells would identify known and previously uncharacterized support cell populations. In addition, as hair cells are continuously replaced, we aimed to identify amplifying and differentiating support cells at different stages of differentiation. We isolated neuromast cells by fluorescence activated cell sorting (FACS) from 5 day post-fertilization (dpf) dissociated transgenic zebrafish in which hair cells, as well as support cells are GFP-positive (*Et(Gw57a);Tg(pou4f3)*; *Figure 1A*) and performed scRNA-Seq analyses using the 10X Chromium System (*Supplementary file 1*). The lateral line also possesses neuromasts with an epithelial planar cell polarity and corresponding gene expression pattern that is offset by 90° depending on which primordium they originated from (primI or primII, (*López-Schier et al., 2004*)). The scRNA-Seq analysis contains cells from all of these neuromasts. For clarity we only discuss and illustrate primI-derived neuromasts.

Unsupervised clustering partitioned 1521 neuromast cells into 14 different clusters (*Butler et al., 2018*). We combined some of the less well-defined clusters and identified seven major neuromast cell types (*Figure 1C–D*). For each population we identified genes specifically expressed or highly enriched (*Figure 1C,E*; *Supplementary file 2*). Dissociating tissues has the caveat that it likely triggers gene expression changes due to loss of adhesion molecules or to a global injury response. We controlled for a global gene expression response by only analyzing genes that are variable between clusters, however cluster-specific gene up- or down regulation can only be controlled for by performing in situ hybridization in intact organs.

The t-distributed Stochastic Neighbor Embedding (t-SNE) plots for genes listed in *Figure 1E* are shown in *Supplementary file 3*. Based on marker gene expression, clusters 1, 2 and 4 encompass the hair cell lineage, with cluster one being differentiated hair cells, cluster two being young hair cells, and four representing proliferating hair cell progenitors (*Figure 1D–E*). The other cycling cells belong to clusters 3 and 10, and because they fail to express hair cell lineage genes, they likely represent amplifying support cells. The proposed hair cell and amplifying support cell lineages are described in detail below. Cells in all other clusters represent different support cell populations (all data can be queried at https://piotrowskilab.shinyapps.io/neuromast_homeostasis_scrnaseq_2018/; see Materials and methods).

To determine if the distinct cell clusters defined by scRNA-Seq can be detected in neuromasts, we performed in situ hybridization experiments with cluster marker genes (*Figure 1E–Q*). Mature, differentiated hair cells are centrally and apically located in a neuromast (*tekt3*, *Figure 1F,R,S*; dark green). Immediately above the mature hair cells are young hair cells that form a ring and express *atoh1b* (cluster 2, *Figure 1G,R,S*). *Figure 1H* shows that *delta* ligands are only expressed in a subset of the young hair cells (light green). *lfng* and *ebf3a* mark the most basal, central support cells (*Figure 1I,J,S,U*; blue). *lfng* is also expressed in support cells that are situated underneath hair cells in the mouse cochlea (*Maass et al., 2016*). The central cell population in neuromasts expresses *gata2a/b* and *slc1a3a/glasta*, which are markers for hematopoietic and neural stem cells, respectively (Figure 3I; *Supplementary file 6*, *Hewitt et al., 2016*; *Llorens-Bobadilla et al., 2015*). Within the central cell cluster, a subset of cells expresses other stem cell-associated genes, such as *isl1* and *fabp7a* (clusters 7, 9; *Figure 1K*; *Kim et al., 2016*; *Makarev and Gorivodsky, 2014*; *Morihiro et al., 2013*; *Shin et al., 2007*). In addition, members of the retinoic acid pathway, such as *crabp2a*, *dhrs3a* and *rdh10a* are restricted to clusters 7 and 9 (*Figure 1E*). Even though central cells express genes characteristic for stem cells in other systems, our lineage tracing experiments showed that they only give rise to hair cells and do not self-renew (*Romero-Carvajal et al., 2015*).

Cells in the D/V poles of neuromasts that express *wnt2* are located immediately adjacent to the mantle cells and proliferate to generate more support cells that do not differentiate into hair cells (see below; *Romero-Carvajal et al., 2015*). As these cells self-renew and possibly represent a stem cell population, we were particularly interested in characterizing new markers for these cells and tested the expression of *sost*, *fsta*, *srrt/ars*, *six2b* and *adcyap1b* (*Figure 1E,L,T*; orange cells). However, all of these polar genes are expressed in more cells than just the ones immediately adjacent to mantle cells, precluding us from obtaining a specific marker for the amplifying cells (*Figure 1T*; red cells). Moreover, D/V polar cells do not form their own cluster but are distributed throughout several

clusters including some mantle cells (clusters 5, 6), central cells (cluster 11) and amplifying, non-differentiating cells (cluster 3).

The most distinct support cell population are the mantle cells represented in clusters 5 and 6 and marked by *ovgp1* and *sfrp1a* (*Figure 1M–N*). Mantle cells are the outermost cells in a neuromast and sit immediately adjacent to amplifying support cells (*Figures 1B,M,N,R–U*). Lineage tracing of mantle cells in medaka revealed that mantle cells give rise to support and hair cells and constitute long term stem cells (*Seleit et al., 2017*). In addition, they give rise to postembryonic neuromasts during development and restore neuromasts on regenerating tail tips (*Dufourcq et al., 2006*; *Ghysen and Dambly-Chaudière, 2007*; *Jones and Corwin, 1993*; *Ledent, 2002*; *Wada et al., 2010*). In addition to representing stem cells, mantle cells may provide the amplifying support cells with niche factors.

We also identified a number of genes that are expressed in a ring-like pattern, such as *fndc7rs4*, *tfap2a*, *tcf7l2*, *hopx*, *cmah* and *alpl* (*Figure 1O*, data not shown). These genes are not restricted to any cluster but are expressed in mantle cells, anterior-posterior (A/P) cells and polar cells. Expression is relatively low in central cells and absent in the hair cell lineage (*Supplementary file 3*). Interestingly, *hopx*, *cmah* and *alpl* are stem cell markers in different systems raising the possibility that they also mark stem cells in neuromasts (*Fuchs, 2009*; *Takeda et al., 2011*). In summary we identified and validated the presence of previously unknown support cell populations, some of which are signaling to trigger regeneration, as shown below.

## Cycling cells characterize the amplifying and differentiating lineages

As proliferation is the basis for zebrafish hair cell regeneration we were particularly interested in identifying cycling support cells. Cells in clusters 10, 3 and 4 express *pcna* (proliferating cell nuclear antigen), required for DNA replication and repair, as well as the mitotic spindle regulator *stmn1a* (*Figures 1D*, *2A and D*; *Supplementary file 4*; *Rubin and Atweh, 2004*). Genes that regulate early versus later phases of the cell cycle are expressed in complementary subsets of the *pcna* +cells (*Figure 2B,C*, *Supplementary file 5*). Genes with the associated GO terms DNA replication and DNA repair, such as *mcm4* are expressed in cells closer to clusters 11 and 14, whereas genes involved in chromosome segregation and mitosis are expressed in cells closer to the younger hair cells in the t-SNE plot (*cdk1*; *Figure 2B,C and D*).

Differential gene expression analysis between the proliferating cells in clusters 3, 4 and 10 revealed that only cluster 4 cells express genes characteristic for the hair cell lineage such as *atoh1b* and *dld* (*Figure 1G–H*; *Cai and Groves, 2015*). As cluster 10 and 3 cells are only defined by the presence of cell cycle genes, we wondered which non-cycling support cell type they might be most closely related to. To mitigate the effect of cell cycle genes, we regressed out S and G2/M phase genes using Seurat's cell cycle scoring function (*Butler et al., 2018*). Using the original cluster classification on the newly generated t-SNE plot, we observed that several cluster 3 (red) cells are now intermingled with D/V support cells in clusters 10, 11, 12 (*Figure 2—figure supplement 1*). Likewise, some of the cluster four hair cell progenitor cells now localize within the central support cells (*Figure 2—figure supplement 1*). Thus, cluster 3 cells likely belong to the cluster of amplifying support cells adjacent to mantle cells that give rise to two undifferentiated daughter cells (*Figure 2E–G*; support cells, red), whereas cluster 4 cells are more central support cells that give rise to two daughters that differentiate into hair cells (*Figure 2E–G*; green cells, *Romero-Carvajal et al., 2015*).

Long term stem cells are often relatively quiescent in the absence of a severe or prolonged injury. We labeled 5dpf homeostatic neuromasts for 24 hrs with BrdU and subsequently scored non-proliferating support cells (grey squares), and progeny of the dividing cells that differentiated into GFP +hair cells (green squares) or remained support cells (red squares, *Figure 2G*; reanalyzed data from *Romero-Carvajal et al., 2015*). To visualize and compute the ratio of quiescent cells, we plotted the location of each progeny and calculated the BrdU index and spatial distribution of the different cell types (*Romero-Carvajal et al., 2015*; *Venero Galanternik et al., 2016*). Amplifying divisions are restricted to the D/V poles, whereas differentiating divisions are more centrally located but do not show a bias toward any quadrant. D/V poles possess more cells than the A/P poles (*Figure 2H*), however, 6.5% of the D/V cells proliferate, whereas only 0–1% proliferate in the A/P poles (*Figure 2I*). Thus, cells in the A/P poles and central cells beneath the hair cells are relatively quiescent during homeostasis (*Cruz et al., 2015*; *Romero-Carvajal et al., 2015*). The expression of the *zona pellucida-like domain-containing protein one* gene *si:ch73-261i21.5* in the A/P poles has a

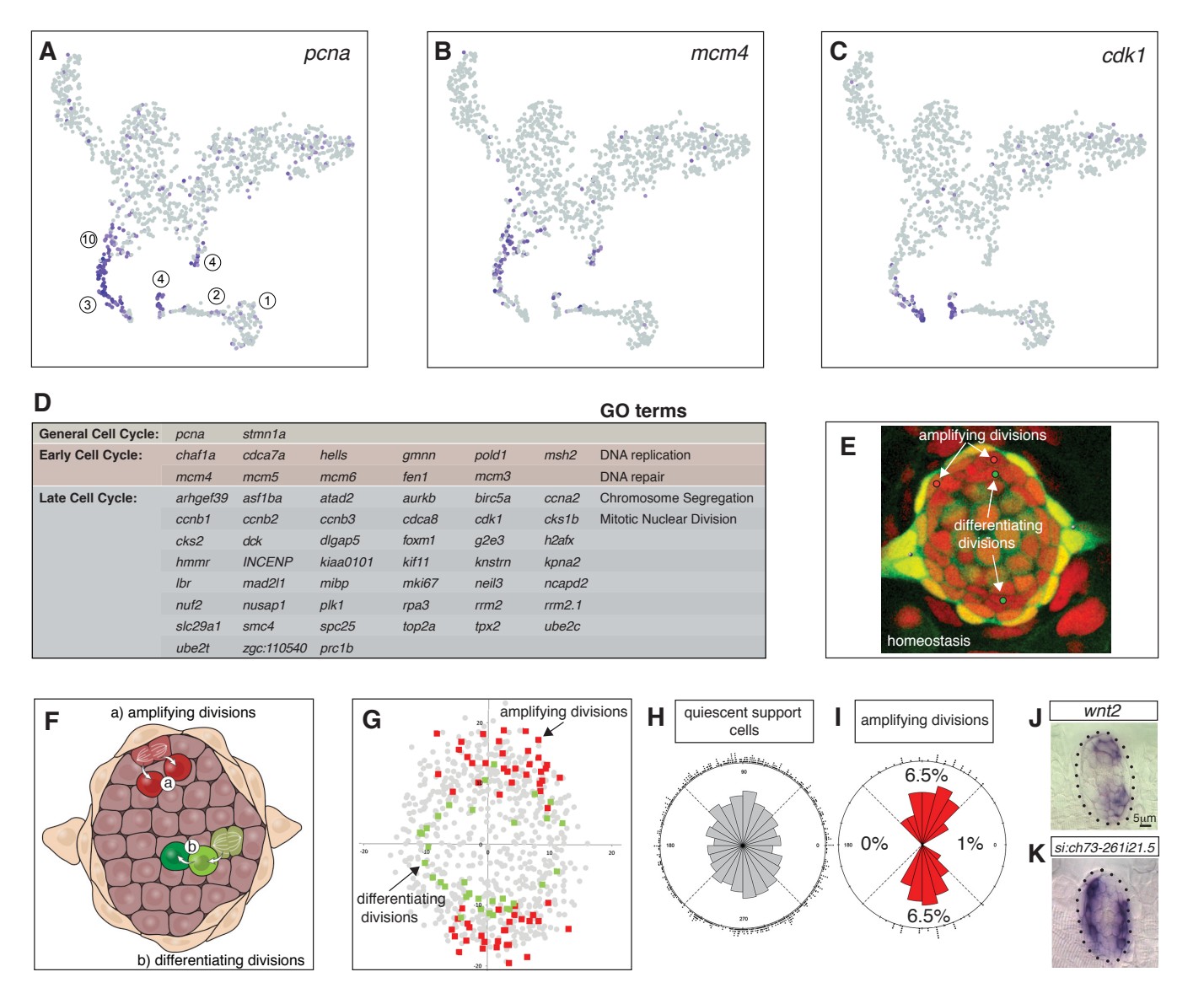

**Figure 2.** Cycling cells characterize the amplifying and differentiating lineages. (A) *pcna* labels all proliferating cells. (B) *mcm4* labels cells that are replicating DNA early in the cell cycle. (C) *cdk1* labels cells in late stages of the cell cycle. (D) Table of early and late cell cycle genes. (E) Still image of a time lapse video of a homeostatic neuromast in which all dividing cells were tracked. Red dots indicate the position of pre-division amplifying cells, green dots indicate differentiating cells (*Romero-Carvajal et al., 2015*). (F) Schematic dorsal view of a neuromast showing that amplifying cells are next to mantle cells in the poles (red), whereas differentiating cells are centrally located (adapted from *Romero-Carvajal et al., 2015*). (G) BrdU analysis of 18 homeostatic neuromasts that were labeled with BrdU for 24 hrs (*Romero-Carvajal et al., 2015*). The position of each dividing support cell was plotted. Cells that divided symmetrically and self-renewed are plotted in red; dividing cells that differentiated into hair cells are in green. Quiescent support cells are in grey and show that cells in the A/P poles are relatively quiescent and if they divide, they differentiate. Mantle cells are not shown. (H, I) Rose diagrams show that the D/V poles possess slightly more cells than the A/P poles (H), however a larger percentage of them proliferates (I). (J) *wnt2* is expressed in the domain of amplifying cells. (K) the *zona pellucida-like domain-containing protein 1gene si:ch73-261i21.5* is expressed in the quiescent region.

DOI: https://doi.org/10.7554/eLife.44431.004

The following figure supplements are available for figure 2:

**Figure supplement 1.** t-SNE plot, cell cycle genes regressed out.
DOI: https://doi.org/10.7554/eLife.44431.005

**Figure supplement 2.** Heatmap of human deafness genes that are expressed in homeostatic lateral line scRNA-Seq data.
DOI: https://doi.org/10.7554/eLife.44431.006

striking complementary expression pattern to the D/V maker *wnt2* and is expressed in quiescent cells (*Figure 2J–K*). Therefore, A/P genes could play a role in regulating quiescence.

Having established different support and hair cell populations and their proliferation status allows us to interrogate the expression pattern of any gene (see web app: https://piotrowskilab.shinyapps.io/neuromast_homeostasis_scrnaseq_2018/ ). For example, a heatmap of human deafness genes demonstrates which genes are expressed in hair cells and which ones are expressed in support cells, crucial information for elucidating their function (*Figure 2—figure supplement 2*; *Supplementary file 6*).

## Inter-cluster relationships reveal how gene networks change in different lineages

Cell cycle analyses suggested the existence of two lineages of cycling cells that possibly represent the amplifying and differentiating support cells identified in time-lapse analyses (*Romero-Carvajal et al., 2015*). To identify possible lineage progression and relationships between non-cycling, cycling and differentiating cells, we generated a dendrogram of the 14 cell clusters using hierarchical clustering (*Figure 3A* and *Figure 1D*, see Materials and methods) with nodes and terminal branches represented by a number. We then produced heatmaps of genes enriched in each of the nodes and branches (*Supplementary file 7*, *8*). Node 15 distinguishes the transcriptome of differentiated hair cells from support cells. In other species, SoxB1 genes characterize the prosensory domain from which hair cells and support cells arise (*Dabdoub et al., 2008*; *Kuzmichev et al., 2012*). Node 17 shows that the SoxB1 family member *sox2*, its target *sox21a* and *sox3* are expressed in support and mantle cells but are downregulated in differentiated hair cells (*Figure 3D*; *Supplementary file 8*, *Supplementary file 3*). Thus, mantle and all support cells constitute the prosensory domain in a lateral line neuromast.

Genes that span multiple support cell clusters identified two lineages emerging within the prosensory domain (*Figure 3B–C*, *Supplementary file 8*). Clusters 7, 9, 8, 14 and 4 comprise the differentiating hair cell lineage, whereas clusters 5, 6, 12, 11, 10 and 3 encompass the amplifying lineage (*Figure 3B–C*). Indeed, lineage tracing experiments determined that central support cells give rise to hair cells (*Romero-Carvajal et al., 2015*), whereas mantle cells give rise to support and hair cells if traced for several months and constitute long term stem cells (*Seleit et al., 2017*).

Heatmaps of factors involved in ribosome and protein synthesis also provide lineage information. Quiescent neural stem cells show low levels of ribosomal subunits and protein synthesis (*Llorens-Bobadilla et al., 2015*). Likewise, we observed mantle (clusters 5, 6) and A/P cells (cluster 13) expressing low levels of *rpl* (*ribosomal protein-L*) and *rps* (*ribosomal protein small subunit*) genes but these levels significantly increase in clusters 12 and 11 and the dividing cells in clusters 10 and 3 (*Figure 3B'* and *Figure 3—figure supplement 1*). Also, the hair cell lineage and the central cells (clusters 7, 8, 9 and 14) show little ribosome synthesis, which drastically increases in dividing hair cell progenitors (*Figure 3C'*, *Figure 3—figure supplement 1*). The low ribosome synthesis levels suggest that central support and mantle cells resemble quiescent stem or progenitor cells.

Mantle cell genes show fairly specific gene expression, such as *tspan1* but also share genes with clusters 12, 13, and amplifying support cells in clusters 10, 3, suggesting a lineage relationship (amplifying lineage; *Figure 3B,E*; *Supplementary file 8*, nodes 18, 23). In addition, clusters 6, 12, 11, 10 and 3 express the D/V polar genes *sost*, *wnt2*, *adcyap1b* and *fsta* (*Figure 3B,F*; *Supplementary file 8*, node 24). These genes label the D/V compartments of neuromasts in which amplifying support cells reside next to mantle cells (*Figure 2E–F*; *Romero-Carvajal et al., 2015*). However, when *sost*+ cells are displaced towards the center of the neuromasts, they downregulate *sost* and differentiate into hair cells (lineage from clusters 11/14 to 4, green arrow in *Figure 3B and G*). The amplifying lineage is also supported by genes such as the pluripotency marker *hopx*, expressed in mantle and A/P cells as well as proliferating, non-differentiating cells in clusters 10 and 3 (*Figures 1E,O* and *3B,G*; *Supplementary file 8*, nodes 18, 23, 3; *Li et al., 2015a*). We conclude that the cells adjacent to mantle cells (and possibly mantle cells themselves) constitute the amplifying lineage.

The differentiating lineage is marked by *atoh1a* which specifies hair cells together with the downstream *delta* ligands (*Figures 3H* and *4A*). *atoh1a* is expressed in non-cycling (clusters 7, 8, 9 and 14) and cycling hair cell progenitors (cluster 4), as well as young hair cells (cluster 2). These cells belong to a subset of central support cells that are marked by progenitor markers *slc1a3a/glasta*,

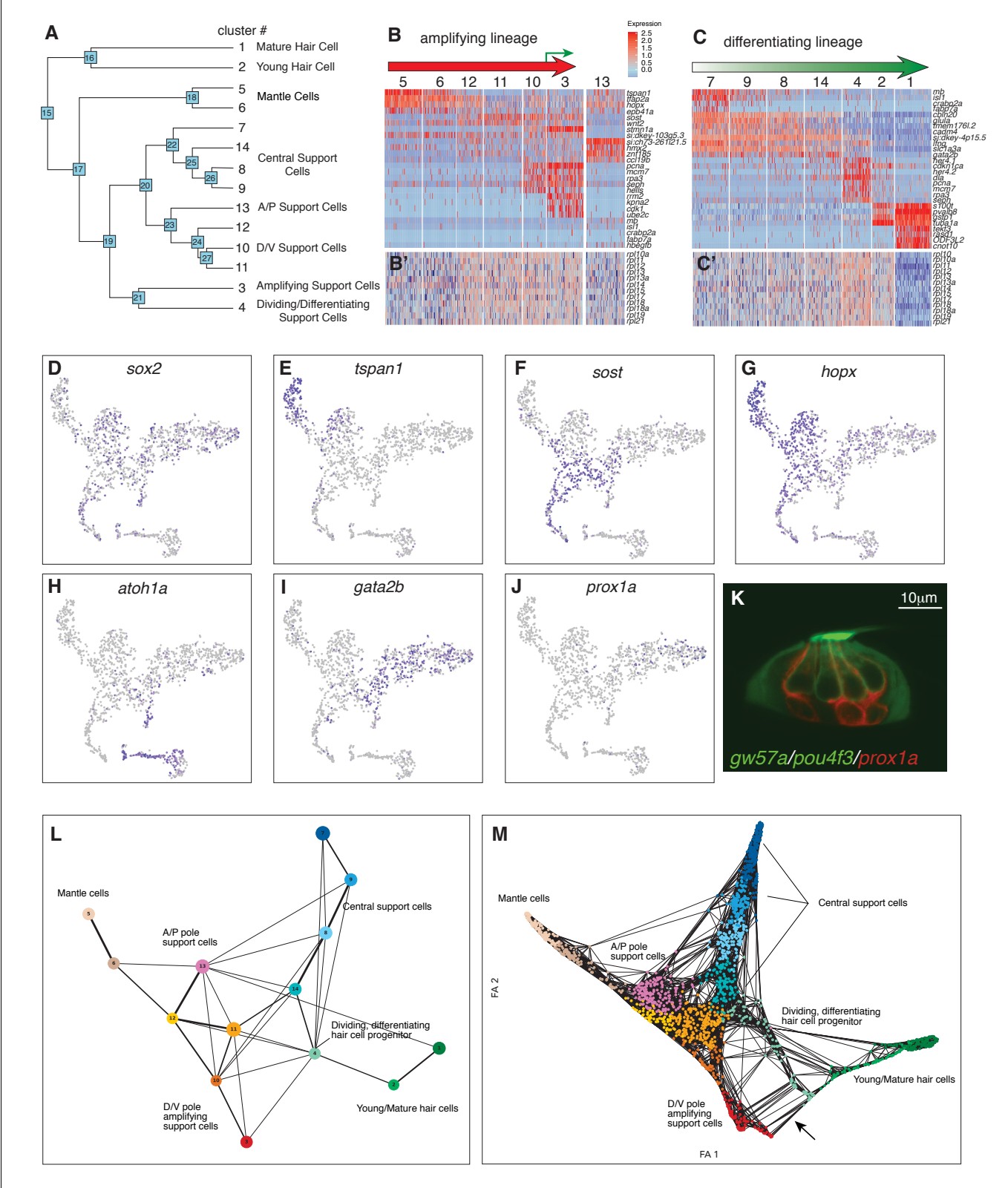

**Figure 3.** Organizing clusters along a putative path of development reveals amplifying and differentiating lineages. (**A**) Dendrogram of cell clusters. Each branch point (node) is labeled. Genes enriched in each branch are shown in **Supplementary file 7**. (**B, C**) Genes selected from the node heatmaps show how some genes are shared between different clusters indicating the existence of two different lineages. Heat map legend shows log2 fold expression changes. (**B**) Amplifying lineage: Mantle cells (clusters 5, 6) to proliferating, self-renewing support cells (cluster 3). Green arrow shows

*Figure 3 continued on next page*

*Figure 3 continued*

that some amplifying cells switch over to the differentiating lineage when displaced toward the center of the neuromast. The relationships with cluster 13 cells is unclear. This cluster is thus set aside. (C) Differentiating lineage: Central support cells (7, 9, 8) to differentiated hair cells (clusters 1, 2). (B', C') Heatmaps of ribosomal genes in the two lineages suggest increased transcription as support cells are activated. Mantle cells, central cells and differentiated hair cells show low levels of ribosome biogenesis (see *Figure 3—figure supplement 1*). (D–J) t-SNE plots of selected genes labeling the different lineages. (D) *sox2* labels all support cells. (E–G) The amplifying lineage is labeled by *tspan1, sost* and *hopx*. (H) *atoh1a* labels the differentiating/hair cell lineage. (I) *gata2b* labels mostly the central cells. (J) *prox1a* labels mostly central cells in cluster 7. (K) A *gw57a; prox1a; pou4f3: GFP* transgenic neuromast shows that red, *prox1a*-positive cells sit immediately beneath the green hair cells. (L) Each node in the PAGA graph represents a cluster and the weight of the lines represents the statistical measure of connectivity between clusters. (M) A ForceAtlas2 plot shows connectivity (KNN, k=15) between individual cells. The arrow points at connections between amplifying cells (cluster 3) and differentiating cells (cluster 4).

DOI: https://doi.org/10.7554/eLife.44431.007

The following figure supplements are available for figure 3:

**Figure supplement 1.** Heatmap of the expression of ribosomal protein genes in homeostatic lateral line cell scRNA-Seq data, reflecting transcriptional activity .
DOI: https://doi.org/10.7554/eLife.44431.008
**Figure supplement 2.** Still images of a video of *prox1a:tagRFP*-positive cells differentiating into *pou4f3:gfp*-positive hair cells.
DOI: https://doi.org/10.7554/eLife.44431.009

*lfng*, *ebf3a*, *gata2b* and *prox1a* (*Figure 1I–J* and *Figure 3I–K*, *Supplementary file 8*, nodes 26, 22 and 7). Indeed, time lapse analyses of regenerating neuromasts in a *prox1a* reporter line show that central cells downregulate *prox1a* as they divide, while turning on the hair cell marker *pou4f3:gfp* (*Figure 3—figure supplement 2*, *Figure 3*-video 1), as described for the mouse cochlea (*Bermingham-McDonogh et al., 2006*). We conclude that central cells (clusters 7, 9, 8,14 and 4) contain or represent hair cell progenitors.

## Dynamics of gene expression during hair cell differentiation

After establishing that hair cell progenitors reside within central cells and that cluster one represents differentiated hair cells, we ordered these cells along a developmental trajectory (henceforth referred to as pseudotime). To define this trajectory, we generated a graph highlighting the relationships between each cluster using partition-based graph abstraction (PAGA, *Plass et al., 2018*; *Wolf et al., 2018b*). Each node represents a cluster and the weight of the lines represents the statistical measure of connectivity between the nodes. The PAGA model is built on a neighborhood graph of single cells and represents a simplified structure from which a path of differentiation can be inferred. We selected clusters 14, 4, 2,1 to represent a putative path of hair cell differentiation. Cluster 14 cells represent the 'root' cell population and cluster one the terminal population, with clusters 4 and 2 representing intermediate populations. Cells in each of these four populations are ordered according to their distance to cluster 14, which is the basis for cell positions in our pseudotime heatmap (*Figure 4I*). It is important to note that amplifying support cells in clusters 3, 10, and 11 also share a relationship to differentiating support cells. This suggests that amplifying support cells may differentiate if displaced to the center of the neuromast (*Figure 3M*, arrow).

Within the hair cell lineage, gene expression changes progressively from the non-cycling progenitors (cluster 14) to differentiated hair cells (cluster 1), reflecting developmental time (*Figure 4A–H*, *Supplementary file 9* and *10*). As progenitors are exiting the cell cycle they turn on differentiation genes, many of which are shared between clusters 2 and 1 (*Supplementary file 3* and *8*, node 16). However, hair cells are subdivided into younger hair cells (cluster 2) and mature hair cells based on differential gene expression (cluster 1; *Supplementary file 3*, nodes 1 and 2), but the younger hair cells in cluster two possess cilia and can be killed with neomycin (*Figure 1R–S*; *Figure 4—figure supplement 1*).

To visualize the expression dynamics of all detected hair cell lineage genes we generated a heatmap in which cells are ordered along pseudotime on the x-axis (*Figure 4I*). This heatmap reveals clusters of genes that possess similar expression dynamics and likely form a regulatory network within each cell cluster (*Figure 4I*). This map of progressive gene activation serves as a blueprint for hair cell specification and differentiation.

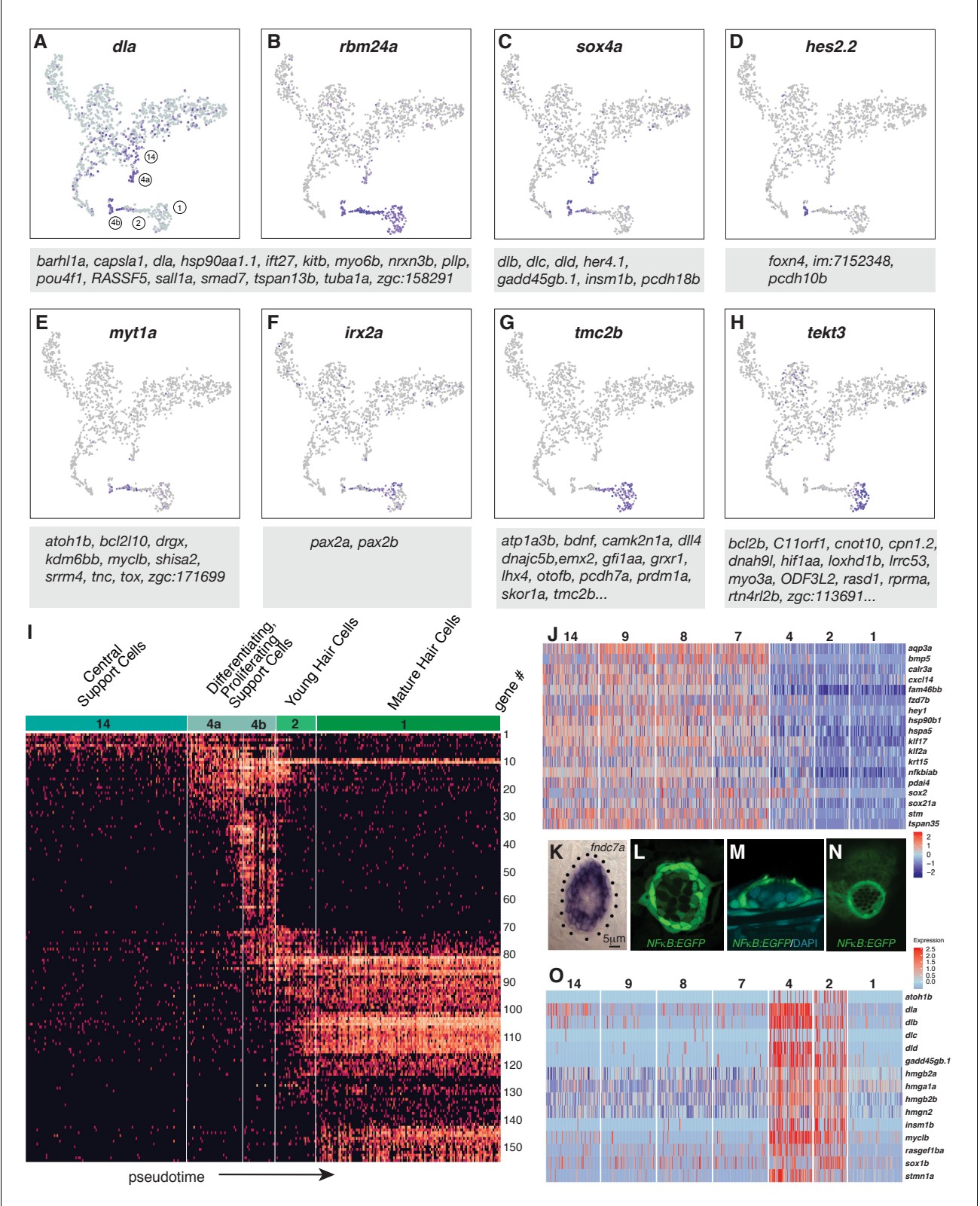

**Figure 4.** Dynamics of gene expression during hair cell differentiation. (A–H) t-SNE plots of selected genes at different stages along the hair cell trajectory. Genes that show similar expression patterns are listed below the respective t-SNE plots. (I) Expression heatmap of hair cell lineage genes with cells from clusters 14, 4a/b, 2, and one ordered along pseudotime (see *Figure 4A*). Each gene # corresponds to the row index in *Supplementary file 11*. (J) Heatmap of genes that are downregulated as hair cells mature. Heat bar shows log2 fold expression changes. (K) In situ of

*Figure 4 continued on next page*

*Figure 4 continued*

*fndc7a*. Genes that are expressed in support cells but are downregulated in the hair cell lineage form a ring of expression. (**L–N**) A *NFκB2:gfp* reporter line shows expression in all support cells but not the hair cell lineage. (**O**) Maturing hair cells downregulated a number of genes as they develop from young to mature hair cells (clusters 2 and 1). Heat bar shows log2 fold expression changes.

DOI: https://doi.org/10.7554/eLife.44431.010

The following figure supplements are available for figure 4:

**Figure supplement 1.** Cilia gene expression in young (cluster 2) and mature hair cell (cluster 1).

DOI: https://doi.org/10.7554/eLife.44431.011

**Figure supplement 2.** Related to *Figure 4*: GO terms of genes down regulated in Hair Cell clusters 1 and 2.

DOI: https://doi.org/10.7554/eLife.44431.012

**Figure supplement 3.** Notch signaling plays an essential role in inhibiting proliferation and maintaining support cell fates.

DOI: https://doi.org/10.7554/eLife.44431.013

Hair cell specification and differentiation also depends on the downregulation of genes (*Matern et al., 2018*). Node 17 (*Supplementary file 8*) shows such genes that are enriched in support cell types and are downregulated in the hair cell lineage (clusters 4, 2 and 1; *Figure 4J*). For example, in situ hybridization with *fndc7a* shows that it labels support cells as in the mouse and that the more apically located young and mature hair cells are unlabeled (*Figure 4K*; *Maass et al., 2016*). Likewise, a *Tg(NFκB:EGFP)* reporter line shows that the *NFκB* pathway that regulates proliferation and self-renewal in other systems is expressed in support cells but that hair cells are GFP-negative (*Figure 4L–N*; *Kanther et al., 2011*; *Rinkenbaugh and Baldwin, 2016*; *Zakaria et al., 2018*). A GO term analysis reveals that genes associated with regulation of transcription, translation, protein folding, cell cycle and Wnt signaling are downregulated in clusters 1 and 2 (*Figure 4—figure supplement 2*). Another group of genes is downregulated as hair cells develop from young to mature hair cells (*Figure 4O*, *Supplementary file 8*, node 2 and 24). These downregulated genes are associated with the GO terms translation, regulation of cell cycle and Notch signaling (*dla, dlb, dlc, dld*). Notch signaling plays an essential role in specifying hair cells versus support cells and a detailed expression analysis of Notch receptors, ligands and downstream targets is shown in *Figure 4—figure supplement 3*. In sum, the pseudotime heatmap provides a blueprint for the succession of gene activation and repression that occurs in support cells as they differentiate into hair cells, thus providing a framework for experimentally inducing hair cell differentiation in mammals.

## scRNA-seq reveals that loss of *fgf3* in central support cells leads to increased proliferation and regeneration

Our previous bulk RNA-Seq analysis of regenerating neuromasts revealed that Fgf pathway genes are downregulated 1 hr after hair cell death ((*Jiang et al., 2014*; *Figure 5A–H'*) suggesting that the downregulation of Fgf signaling could be involved in triggering regeneration, as was shown for Notch signaling (*Ma et al., 2008*; *Romero-Carvajal et al., 2015*). Indeed, hair cell regeneration is enhanced in *fgf3* mutant larvae and even during homeostasis the total cell number per neuromast is increased (*Figure 5I–L*).

Because *fgf3* disappears as hair cells die, we wondered if *fgf3* is expressed in hair cells. The scRNA-Seq analysis of homeostatic neuromasts shows that ligand and receptor expression is complex, and that Fgf signaling is not active in young or mature hair cells (*Figure 5M*; clusters 2, 1). *fgf3* is expressed exclusively in central support cells (clusters 7, 8, 9) and is downregulated in response to death of the overlying hair cells (*Figure 5A'*). We generated a *fgf3* knock-in line that recapitulates *fgf3* expression (*Figure 5N–O*, *Figure 5*-video 1) and a cross with a *prox1a* reporter line shows that *fgf3* and *prox1a* are co-expressed, as predicted by the scRNA-Seq data (*Figure 5P–Q*). To understand the cellular basis of the increased regeneration response, we performed BrdU analyses of homeostatic and regenerating *fgf3*-deficient neuromasts. During both homeostasis and regeneration, proliferation is significantly increased in *fgf3*$^{-/-}$ neuromasts (*Figure 6A–F*). Downregulation of Fgf signaling by expression of *dn:fgfr1a* during homeostasis also increases proliferation and neuromasts are significantly bigger in *fgfr1a/fgfr2* double mutants, similarly to *fgf3*$^{-/-}$ (*Figure 6—figure supplement 1*, *Figure 6G–L*, *Figure 5L*). Therefore, *fgf3* likely signals via *fgfr1a* and *fgfr2* receptors. The BrdU plots and the rose diagrams show that in *fgf3*$^{-/-}$ homeostatic and regenerating neuromasts amplifying divisions are not restricted to the D/V poles (*Figure 6B–B'* and *D–D'*). This pattern of

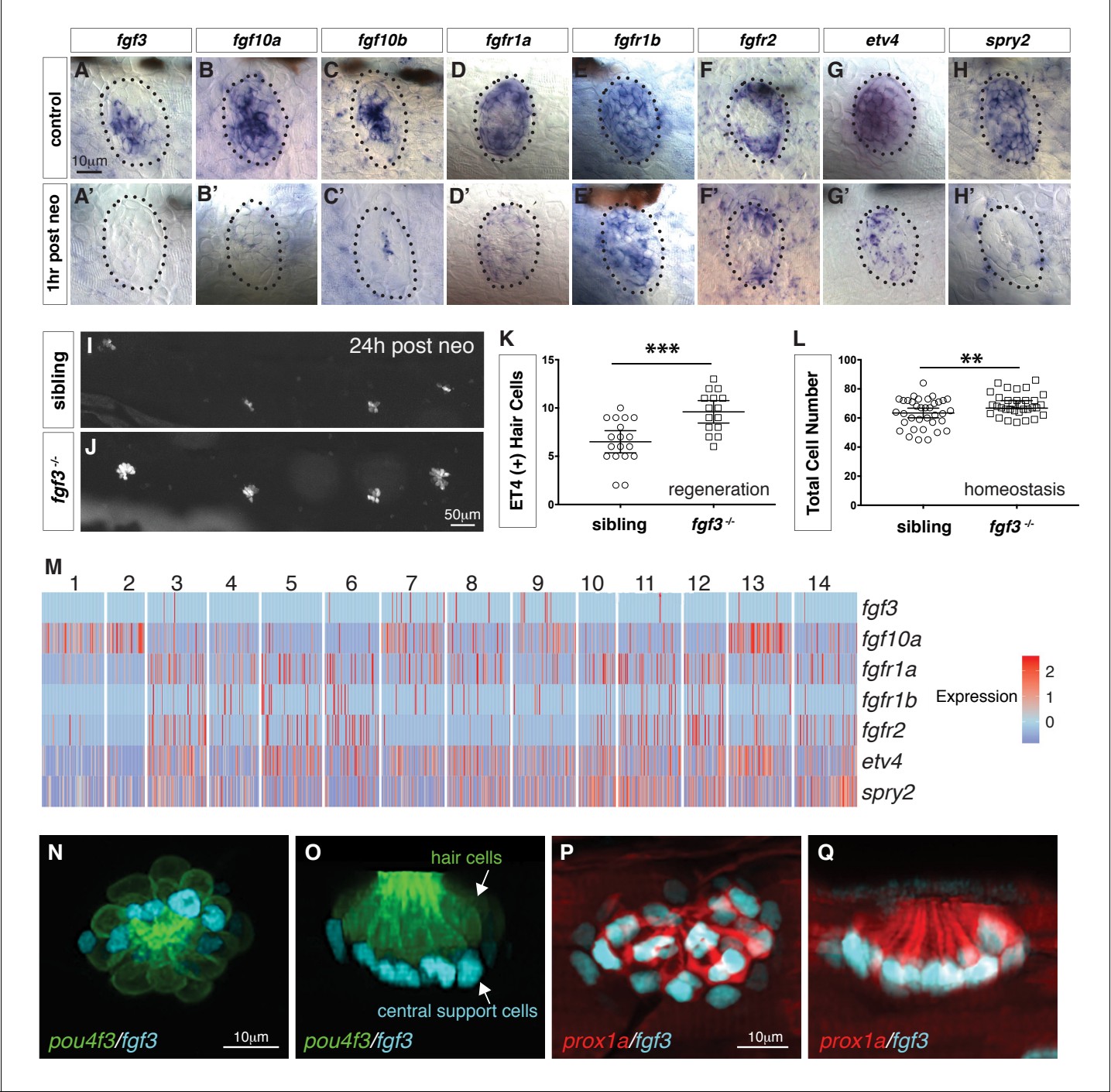

**Figure 5.** scRNA-seq reveals that *fgf3* is expressed in central support cells and its downregulation enhances regeneration. (A–H) Fgf pathway genes are expressed in 5dpf neuromasts. (A'–H') Fgf pathway genes are downregulated 1 hr after hair cell death (post neo). (I and J) DASPEI staining of sibling (I) and *fgf3*−/− larvae (J) 24 hrs post neomycin showing hair cells. (K) Quantification of *ET4:GFP* (+) hair cells 24 hrs post neomycin in siblings and *fgf3*−/− neuromasts. *fgf3*−/− show increased hair cell and support cell numbers during regeneration. Error bars show the 95% confidence interval (CI). ***p<0.0004, unpaired t-test. (L) Quantification of total neuromast cell numbers at 5dpf in homeostatic sibling and *fgf3*−/− neuromasts. Even during homeostasis *fgf3*−/− neuromasts possess more cells. Error bars show 95% CI. **p<0.0084, unpaired t-test. (M) Homeostasis scRNA-Seq expression heatmap of Fgf pathway genes. Heat bar shows log2 fold expression changes. (N–O) Double transgenic for *pou4f3:gfp* and *fgf3:h2b-mturquoise2* at 5dpf. Dorsal and lateral views, respectively. Hair cells are in green, *fgf3*-expressing nuclei of central cells are in cyan. (P-Q) Double transgenic for *prox1a:tag-rfp* and *fgf3:h2b-mturquoise2* at 5dpf. *prox1a* and *fgf3* are co-expressed in central support cells.

DOI: https://doi.org/10.7554/eLife.44431.014

*Figure 5 continued on next page*

*Figure 5 continued*

The following video is available for figure 5:

**Figure 5—video 1.** 3D animation of a *fgf3:H2B-mturquoise2* and *pou4f3:gfp*-expressing neuromast.

DOI: https://doi.org/10.7554/eLife.44431.015

proliferating cells looks almost identical to the ones observed after upregulation of Wnt signaling (*Romero-Carvajal et al., 2015*).

## *fgf3* downregulation leads to Wnt-induced proliferation in parallel to Notch signaling

To identify genes/pathways underlying the increased proliferation in *fgf3*[-/-], we first performed bulk RNA-Seq analysis with five dpf homeostatic *fgf3*[-/-] and siblings. However, the differences in gene expression between siblings and mutants was too low. We therefore performed scRNA-Seq analyses on 1459 *fgf3* mutant and 1932 sibling cells (*Supplementary file 13*). A t-SNE plot in which we plotted mutant and sibling datasets together demonstrates that the variance between the two datasets is small as the two datasets intermingle (*Figure 7A*, sibling blue, mutant red). The plot also shows that mutant cells contribute to each cluster and that therefore no major cell type is missing in *fgf3*[-/-] (*Figures 7A–B* and *1D*).

However, the scRNA-Seq analysis revealed *fgf3* targets that are down- or upregulated in the mutants (*Figure 7C*, *Figure 7—figure supplement 1*). We were particularly interested in genes that regulate the Wnt pathway and/or proliferation and identified that some of the D/V polar genes, such as *sost* and *adcyap1b* are downregulated in the mutants (*Figure 7C*, *Figure 7—figure supplement 1A*). We validated the downregulation of Wnt inhibitor *sost* by in situ hybridization in *fgf3*[-/-] and *dn:fgfr1* larvae (*Figure 7D–E'*). Interestingly, *sost* is also downregulated 1 hr after hair cell death in wild type larvae, suggesting that the downregulation of Fgf signaling after neomycin could also be responsible for the downregulation of the Wnt inhibitor *sost* (*Figure 7F–F'*). Also, the Wnt target gene *wnt10a* is upregulated, illustrating that Wnt signaling is increased in *fgf3* mutants (*Figure 7G–G'*, *Figure 7—figure supplement 1A*).

To interrogate if the increase in proliferation in Fgf pathway mutants is due to the upregulation of Wnt signaling, we abrogated Wnt signaling in *fgf3*[-/-] by crossing them with *hs:dkk1* fish. *hs:dkk1* significantly inhibits proliferation in siblings and it also reduces proliferation and hair cell numbers in *fgf3*[-/-] (*Figure 8A–F*). Therefore, in homeostatic neuromasts *fgf3* inhibits proliferation by inhibiting Wnt signaling, possibly via *sost*. As Notch signaling also inhibits proliferation via inhibiting Wnt signaling, we wondered if Notch and Fgf act in the same pathway (*Romero-Carvajal et al., 2015*). Fgf signaling does not act upstream of Notch signaling, as Notch pathway members are not affected by in situ hybridization in *fgf3*[-/-] (*Figure 8G–K'*, *Figure 9*). *fgf3* on the other hand, is slightly downregulated after Notch signaling inhibition with a γ-secretase inhibitor (*Figure 8M'*) while *fgf10a* and *fgfr1a* are not affected (*Figure 8O–P'*). It is therefore possible that Notch signaling plays some minor role in inducing/maintaining *fgf3* expression. However, Notch signaling does not inhibit Wnt signaling via the upregulation of Fgf3 signaling, as shown by experiments in which we constitutively activated Notch signaling in *fgf3*[-/-] by expressing the Notch intracellular domain NICD. Activation of NICD in siblings during regeneration strongly inhibits proliferation and hair cell production (*Figure 8Q,R,U,V*, *Romero-Carvajal et al., 2015*; *Wibowo et al., 2011*). Activation of NICD in *fgf3*[-/-] also strongly inhibits proliferation and hair cell numbers demonstrating that Notch signaling does not require Fgf to inhibit Wnt signaling (*Figure 8S–V*).

We conclude from these data that Notch and Fgf signaling largely act in parallel to inhibit Wnt signaling, with a small amount of proliferation being inhibited by Notch via the upregulation of Fgf signaling (*Figure 9*). Thus Notch and Fgf need both to be independently and transiently downregulated for efficient hair cell regeneration.

## Discussion

The scRNA-Seq analyses reported here have characterized in unprecedented detail the different support cell populations present in a homeostatic neuromast. As such, our findings have enabled us to detect subtle gene expression changes in *fgf3*[-/-]. Importantly, as dying hair cells are continuously

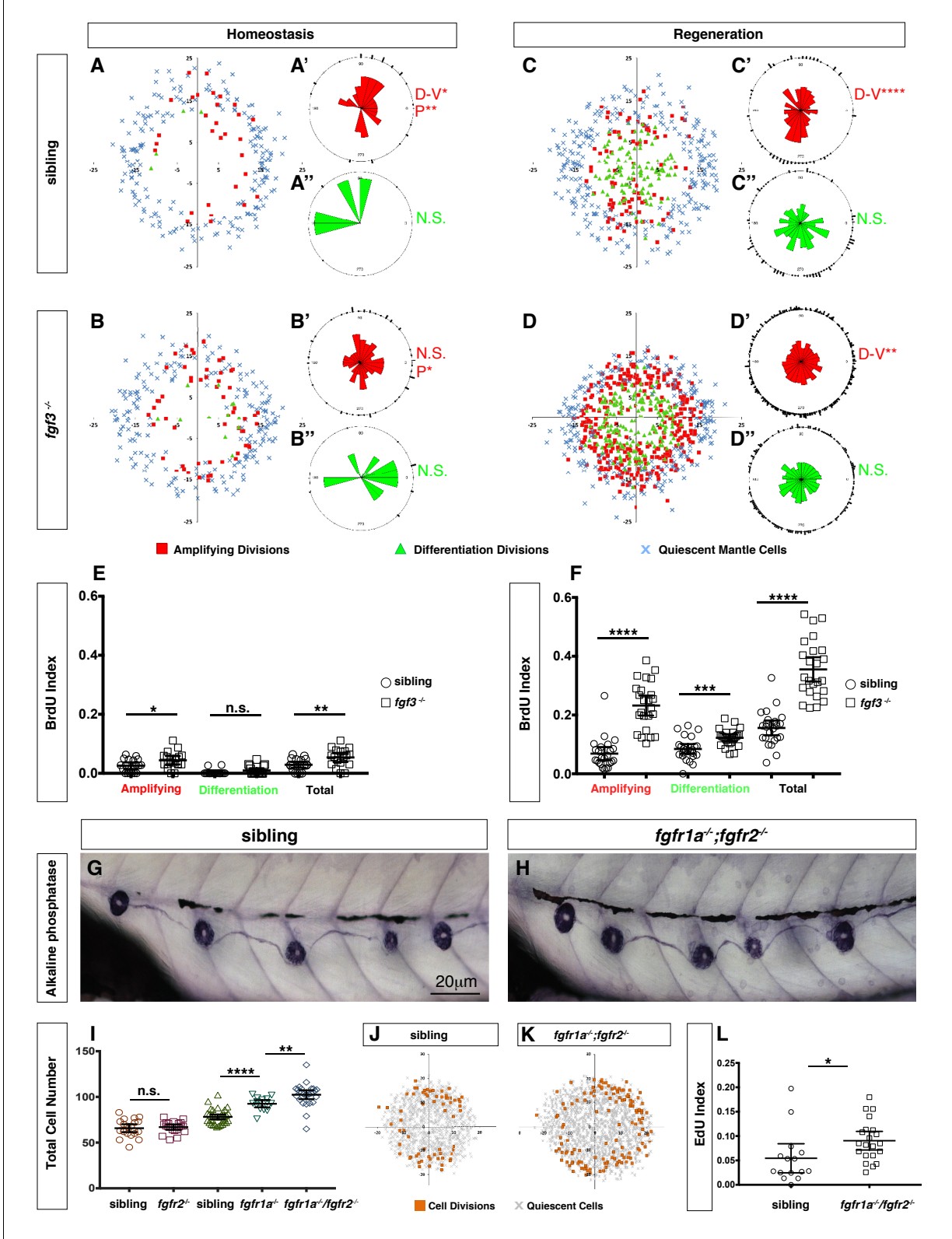

**Figure 6.** Loss of *fgf3*[-/-] causes increased support cell proliferation during homeostasis and regeneration. (A–B) Spatial analysis of amplifying (red squares), differentiation (green triangles) cell divisions and quiescent mantle cells (blue X's) in sibling and *fgf3*[-/-] neuromasts during homeostasis. Quiescent and BrdU-positive cells from 18 neuromasts are superimposed onto the same X-Y plane. N.S. = not significant. (A'–B'') Rose diagrams of the angular positions of BrdU(+) support cells (red) or hair cells (green) in sibling and *fgf3*[-/-] during homeostasis. D/V clustering and directional bias to the
*Figure 6 continued on next page*

*Figure 6 continued*

posterior was analyzed with a Binomial distribution test, *p<0.05, **p<0.008. (**C–D**) Spatial analysis of amplifying and differentiation cell divisions or quiescent mantle cells in sibling and *fgf3*⁻/⁻ 24 hrs post neomycin. (**C'–D''**) Rose diagrams of the angular positions of BrdU-positive support cells or hair cells in sibling or *fgf3*⁻/⁻ 24 hrs post neomycin. D/V clustering and directional bias to the posterior was analyzed with a Binomial distribution test, ****p<0.00001, **p<0.008 (**E**) BrdU index of amplifying, differentiating and total cell divisions in sibling and *fgf3*⁻/⁻ during homeostasis. Error bars show 95% CI. p-value determined by unpaired t-test, *p<0.03, **p<0.007. (**F**) BrdU index of amplifying, differentiating and total cell divisions in siblings and *fgf3*⁻/⁻ mutants during 24 hrs post neomycin treatment. Error bars show 95% CI. p-value determined by unpaired t-test, ***p<0.0005, ****p<0.0001. (**G–H**) Alkaline phosphatase staining of sibling or *fgfr1a*⁻/⁻/*fgfr2*⁻/⁻ at 5dpf. (**I**) Quantification of total neuromast cell number at 5dpf in siblings, *fgfr2*⁻/⁻, *fgfr1a*⁻/⁻ and *fgfr1a*⁻/⁻/*fgfr2*⁻/⁻. Error bars show 95% CI. p-value determined by unpaired t-test, **p<0.007, ****p<0.0001. (**J–K**) Spatial analysis of all cell divisions (orange squares) or quiescent cells (grey X) in sibling or *fgfr1a*/*fgfr2*⁻/⁻ during homeostasis. (**L**) EdU index of total cell divisions in siblings and *fgfr1a*/*fgfr2*⁻/⁻ mutants during homeostasis. Error bars show 95% CI. p-value determined by unpaired t-test, *p=0.03.

DOI: https://doi.org/10.7554/eLife.44431.016

The following figure supplement is available for figure 6:

**Figure supplement 1.** Expression of *dnfgr1* also induces neuromast cell proliferation.

DOI: https://doi.org/10.7554/eLife.44431.017

replaced during homeostasis, our analyses also characterized support cells as they differentiated into hair cells. Even though lineage relationships cannot be inferred from scRNA-Seq data alone, the results of our pseudotime and cell classification delineate lineages that have been experimentally confirmed by time lapse and lineage tracing analyses (*Romero-Carvajal et al., 2015*; *Seleit et al., 2017*; *Viader-Llargués et al., 2018*; *Figure 2E–F* and *Figure 3A–C and L–M*). The majority of hair cells originate from central support cells without bias to any of the poles (*Figure 3M*, clusters 7, 9, 8, 14 and 4), whereas amplifying divisions are strongly biased towards the D/V poles and occur in the periphery adjacent to mantle cells (*Figure 3M*, clusters 10 and 3).

## Cells can transition from an amplifying to differentiating lineage

Interestingly, the PAGA graph (*Figure 3M*) shows a connection between cluster 3 (the amplifying support cells) and cluster 4 (the proliferating differentiating support cells) suggesting that the amplifying support cells can turn on differentiation genes as they are displaced toward the center of the neuromast and also become hair cells. This hypothesis is supported by time lapse movies of amplifying support cells during homeostasis and regeneration (*Romero-Carvajal et al., 2015*). Our finding shows that the amplifying and hair cell lineages are not predetermined, but rather that the location of cells within the neuromast and the signals they are exposed to determine their fate.

## Neuromasts likely possess quiescent and active stem cells

Based on cell behavior and gene expression studies we postulate that lateral line neuromasts possess active and quiescent stem cells. Because amplifying support cells in the D/V poles self-renew continuously and only differentiate if displaced into the center, we postulate that amplifying cells represent active stem cells. In addition, neuromasts possibly possess two populations of quiescent stem cells. The first population are cells in the A/P compartment that are relatively quiescent during homeostasis and regeneration (*Figure 2H*, (*Cruz et al., 2015*; *Romero-Carvajal et al., 2015*). These cells start to proliferate after manipulations of the Fgf, Notch or Wnt pathways (*Figure 6D*, *Romero-Carvajal et al., 2015*). Long term lineage tracing experiments have to be performed to assess if they indeed present stem cells. The second quiescent, long term stem cell population are the mantle cells that give rise to hair cells in long term lineage analyses but that do not respond to acute neomycin-induced hair cell death (*Romero-Carvajal et al., 2015*; *Seleit et al., 2017*; *Viader-Llargués et al., 2018*). Support cells also give rise to mantle cells when mantle cells are ablated (*Viader-Llargués et al., 2018*) arguing that active and quiescent stem cells can convert back and forth as shown in other systems (*Clevers and Watt, 2018*). For example, in the intestine slow-cycling, label-retaining stem cells in the +4 position can give rise to stem cells at the bottom of the crypt, which conversely can also give rise to +4 stem cells (*Takeda et al., 2011*).

Quiescent stem cells in hematopoiesis, the brain, skeletal muscle, hair follicle and Drosophila germline stem cells are also distinguished by their low level of protein synthesis and transcription (*Blanco et al., 2016*; *Llorens-Bobadilla et al., 2015*; *Sanchez et al., 2016*; *Signer et al., 2014*; *Zismanov et al., 2016*). A characteristic of stem cell activation is consequently the upregulation of

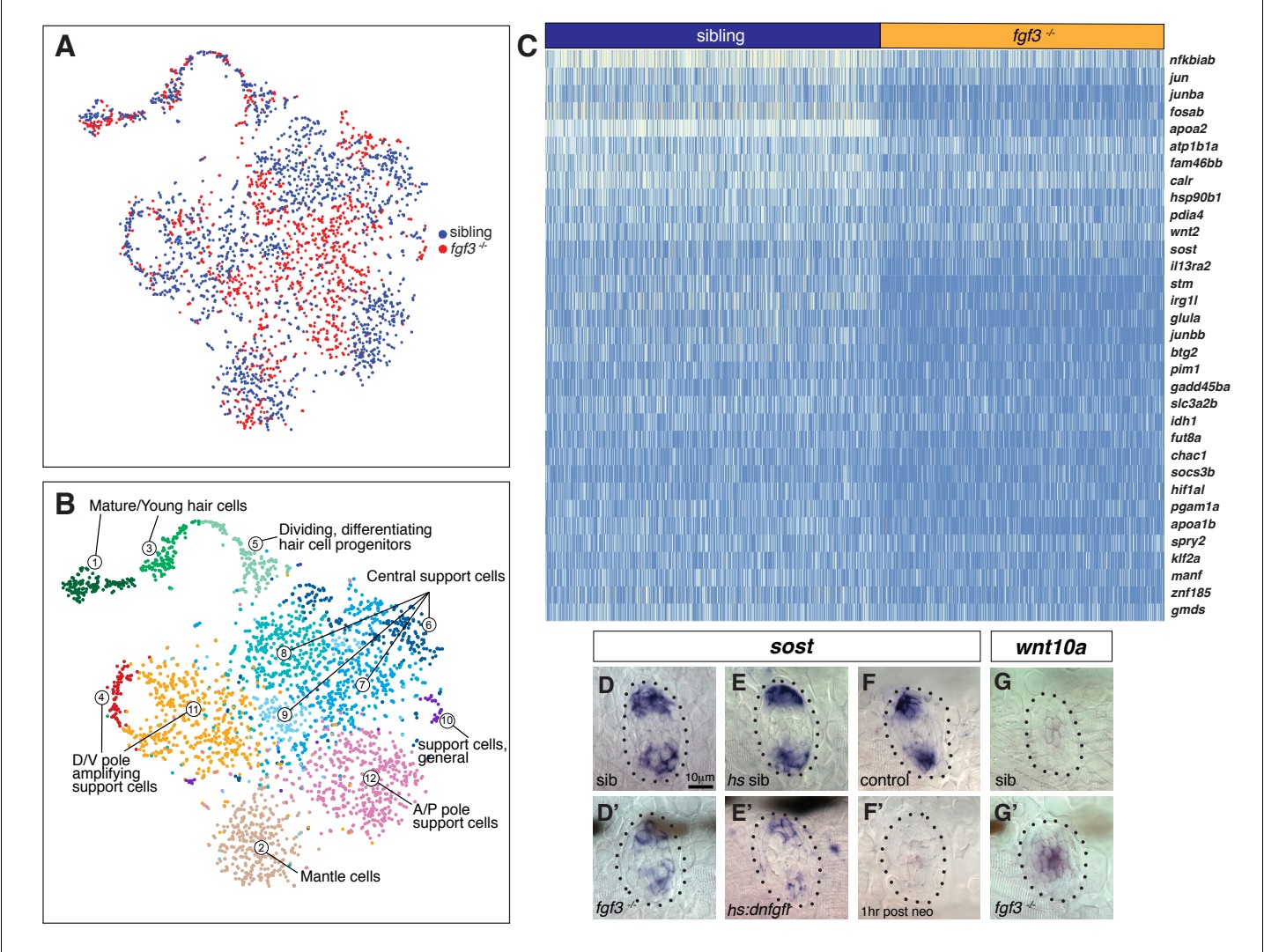

**Figure 7.** scRNA-Seq analysis of *fgf3* mutants identifies *fgf3* targets. (**A**) *t*-SNE plot depicting the integration of *fgf3* mutant and sibling scRNA-Seq data sets. Sibling cells in blue, *fgf3⁻/⁻* cells in red. (**B**) Graph-based clustering of both *fgf3* mutant and sibling data sets. No major cluster is missing in *fgf3⁻/⁻*. (**C**) Heatmap of genes downregulated in five dpf *fgf3⁻/⁻* neuromasts. (**D–F'**) *sost* expression is downregulated in 5dpf *fgf3⁻/⁻* mutants; after expression of *dnfgfr1a* and after 1 hr post neomycin treatment. (**G–G'**) The Wnt target gene, *wnt10a*, is upregulated in 5dpf *fgf3⁻/⁻* neuromasts.

DOI: https://doi.org/10.7554/eLife.44431.018

The following figure supplement is available for figure 7:

**Figure supplement 1.** Selection of genes that are differentially expressed in *fgf3* mutants.

DOI: https://doi.org/10.7554/eLife.44431.019

protein synthesis and associated factors, such as ribosomal proteins (*Baser et al., 2017*; *Signer et al., 2014*). A heatmap of *rpl* genes and *rps* genes shows that in neuromasts these genes are lowly expressed in central support cells (clusters 7, 8, 9, 14), mantle cells (cluster 5, 6) and A/P cells (cluster 13) but are upregulated in dividing cells (clusters 3 and 4) and in D/V cells (clusters 11, 12; *Figure 3B', C', Figure 3—figure supplement 1*). Synthesis of these ribosomal proteins is completely shut down in mature hair cells (cluster 1; *Figure 3C'*). Based on their low ribosomal protein synthesis central support cells and mantle cells resemble quiescent stem cells, whereas D/V cells and dividing cells resemble activated stem cells or progenitor cells.

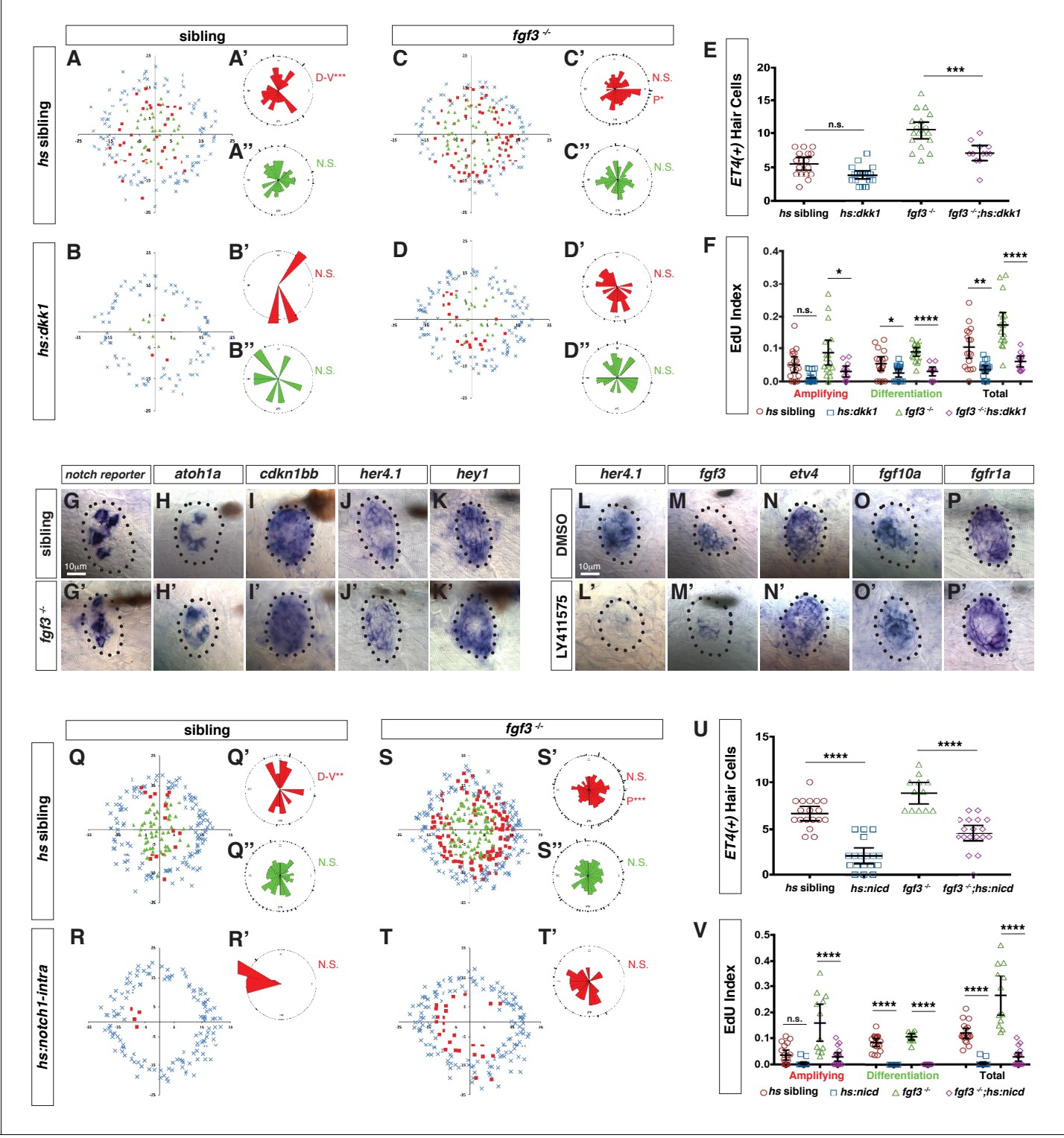

**Figure 8.** Fgf and Notch signaling largely act in parallel to inhibit Wnt-induced proliferation during homeostasis. (A–D") Spatial analysis of amplifying and differentiating cell divisions in sibling and *fgf3⁻/⁻* with or without *dkk1* expression 24 hrs post neomycin. D/V clustering and directional bias to the posterior was analyzed with a Binomial distribution test, *p=0.03, **p<0.008. (E) Quantification of *ET4:GFP*-positive hair cells 24 hrs post neomycin in sibling and *fgf3-/-* with or without *dkk1*. Error bars show 95% CI. p-value determined by Anova and Tukey post-Hoc test, ***p=0.001. (F) EdU index of amplifying, differentiating and total cell divisions in sibling and *fgf3⁻/⁻* with or without *dkk1* 24 hrs post neomycin. Error bars show 95% CI. p-value determined by Anova and Tukey post-Hoc test, *p<0.02, **p<0.004, ****p<0.0001. (G–K') Expression of Notch pathway genes. (G–G') the *notch*

*Figure 8 continued on next page*

*Figure 8 continued*

reporter, *atoh1a* (H–H'), *cdkn1bb* (I–I'), *her4.1* (J–J') and *hey1* (K–K') are unchanged in 5dpf *fgf3⁻/⁻*. (L–P') LY411575 inhibits *her4.1* (L–L') and *fgf3* (M–M'), but not *etv4* (N–N'), *fgf10a* (O–O'), or *fgfr1a* (P–P') in 5dpf neuromasts. (Q–T') Spatial analysis of amplifying and differentiating cell divisions in sibling and *fgf3⁻/⁻* with or without *notch1a-intracellular domain* expression 24 hrs post neomycin. D/V clustering and directional bias to the posterior was analyzed with a Binomial distribution test, **p<0.001, ***p<0.004 (U) Quantification of *ET4:GFP*-positive hair cells 24 hrs post neomycin in sibling and *fgf3⁻/⁻* with or without *notch1a-intracellular domain (nicd)*. Error bars show 95% CI. p-value determined by Anova and Tukey post-Hoc test, ****p<0.0001. (V) Quantification of amplifying, differentiating and total cell divisions in sibling and *fgf3⁻/⁻* with or without *nicd* 24 hrs post neomycin. Error bars show 95% CI. p-value determined by Anova and Tukey post-Hoc test, ****p<0.0001.
DOI: https://doi.org/10.7554/eLife.44431.020

## Support cells share gene expression profiles with stem cells in other organs

The notion that some support cells constitute stem cells is also supported by the finding that their gene expression profiles share a number of genes with stem cells in the CNS, heart, intestine or hair follicles. For example, neural stem cells in the subventricular zone of the CNS and radial glial cells are characterized by glutamate aspartate transporter (GLAST) and Prominin-1 (also known as CD133) expression (*Llorens-Bobadilla et al., 2015*). In neuromasts, *slc1a3a/glasta* and *prominin1a* are expressed strongly in central and A/P support cells and *prominin1b* is specifically enriched in A/P cells. Likewise, genes associated with intestinal, hair follicle and neural stem cells (NSCs), such *hopx* and *alpl* are strongly expressed in mantle, A/P, and amplifying cells adjacent to mantle cells, but are downregulated in central support cells, forming a ring of expression similarly to *fndc7rs4* (*Figure 1O* and *Figure 3G*, *Li et al., 2015a*; *Liu et al., 2018*). Genes specifically expressed in central support cells beneath the hair cells likewise express genes that label stem cells in other organs, such as *fabp7/Blbp*, which labels glioma stem cells, radial glia cells, NSCs, (*Kim et al., 2016*; *Morihiro et al., 2013*) and *isl1* that is expressed in quiescent intestinal stem cells and stem cells in

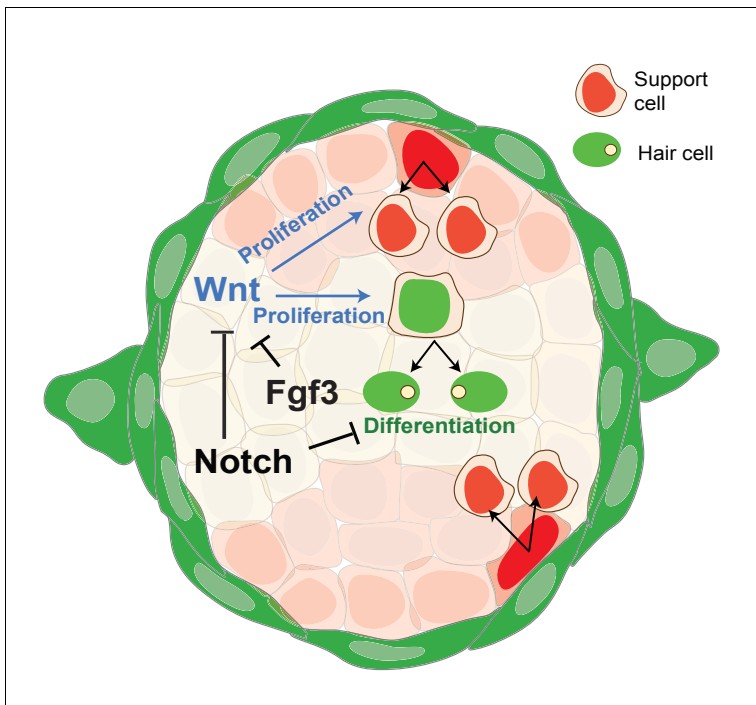

**Figure 9.** Schematic of signaling pathway interactions in a homeostatic neuromast (modified after *Romero-Carvajal et al., 2015*). Notch and *fgf3* inhibit Wnt signaling in parallel in homeostatic neuromasts. The downregulation of Notch and Fgf signaling after hair cell death leads to the upregulation of Wnt signaling and proliferation in central, differentiating and D/V, amplifying support cells.
DOI: https://doi.org/10.7554/eLife.44431.021

the heart (*Makarev and Gorivodsky, 2014*; *Shin et al., 2015*). However, in lateral line neuromasts central cells only give rise to hair cells and we therefore do not consider them to be stem cells.

A comparison of the transcriptional profiles of support cells in regenerating species, such as zebrafish and chicken with mouse support cells will be highly informative. Do mammalian support cells also still express many of the above-mentioned stem cell genes, or do mammalian support cells represent a more differentiated cell population? The results from such analyses will help us determine if mammalian support cells need to be reprogrammed for efficient induction of regeneration.

## *fgf3* inhibits Wnt signaling and proliferation possibly via Sost

Wnt signaling induces and is required for support cell proliferation in neuromasts (*Head et al., 2013*; *Jacques et al., 2014*; *Romero-Carvajal et al., 2015*; *Wada and Kawakami, 2015*). In *fgf3⁻/⁻* Wnt signaling is upregulated causing cells to proliferate (*Figure 8*). We found that of the neuromast-expressed secreted Wnt inhibitors *sfrp1a*, *dkk2, sost* and *sostdc1b,* only *sost* is downregulated in *fgf3⁻/⁻* (*Figure 7D'*, data not shown, *Romero-Carvajal et al., 2015*; *Wada et al., 2013*). Since *sost* is also downregulated after hair cell death (*Figure 7F'*) it may have an important role in inhibiting hair cell regeneration. As Wnt signaling also induces proliferation in mammalian hair cell progenitors (*Chai et al., 2012*; *Jacques et al., 2012*; *Jan et al., 2013*; *Samarajeewa et al., 2018*; *Shi et al., 2012*) a role for Sost paralogs in this system should be tested. According to several data bases, Sost is not, or only lowly expressed in the mammalian ear sensory epithelia, whereas Sostdc1 is robustly expressed.

Fgf signaling could also inhibit proliferation by regulating other receptor tyrosine kinase activities, for example via inhibition of EGFR by Sprouty2 (*Balasooriya et al., 2016*). *sprouty2* is also down regulated in *fgf3⁻/⁻* after hair cell death and could perform a similar inhibitory function in homeostatic neuromasts (*Supplementary file 14*). Fgf signaling also inhibits proliferation in the regenerating utricle and the basilar papilla of chicken (*Jiang et al., 2018*; *Ku et al., 2014*). Likewise, addition of Fgf2 or Fgf20 to auditory or vestibular cultures inhibits support cell proliferation (*Ku et al., 2014*; *Oesterle et al., 2000*), suggesting that the inhibitory effect of Fgf signaling on progenitor proliferation is evolutionary conserved in species that can regenerate their sensory hair cells.

Experiments utilizing chemical inhibition of Fgfr have been less clear and in some studies had no effect on proliferation or even led to the inhibition of proliferation (*Jacques et al., 2012*; *Ku et al., 2014*; *Lee et al., 2005*). The differences in the effect of chemical inhibition can likely be attributed to differences in culture conditions, timing or doses of drug treatment, underscoring the importance of utilizing multiple methods of signaling pathway inhibition, particularly gene mutations.

## A role for fgf in specification?

While our data suggest that the main role of *fgf3* in mature neuromasts is in regulating proliferation, it is also possible that *fgf3* acts to maintain hair cell progenitors in an undifferentiated, non-sensory state as has been observed in the developing zebrafish ear (*Maier and Whitfield, 2014*). Likewise, in the mouse cochlea loss of *Fgfr3* results in increased hair cells at the expense of support cells (*Hayashi et al., 2007*; *Puligilla et al., 2007*), while activating mutations in *Fgfr3* or loss of function mutation in the Fgfr inhibitor *Spry2* lead to transformation of one support cell type into another (*Mansour et al., 2013*; *Mansour et al., 2009*; *Shim et al., 2005*). However, a possible inhibitory effect of *fgf3* on hair cell fate has to be rather subtle, as we only observe a limited increase in hair cell numbers, and our scRNA-Seq analyses did not reveal any obvious candidates that might regulate cell fate in a Fgf-dependent fashion.

## Interactions between the Fgf and Notch pathways

Just like Fgf, Notch signaling is immediately downregulated after hair cell death causing cell proliferation. During homeostasis both pathways inhibit proliferation through negative regulation of Wnt activity (*Ma et al., 2008*; *Romero-Carvajal et al., 2015*). To test if Fgf and Notch act in the same pathway we performed epistasis experiments. In *fgf3⁻/⁻* mutants expression of a Notch reporter and Notch target genes are not affected suggesting that Notch signaling is largely intact in *fgf3⁻/⁻* (*Figure 8G–K'*). Also, while pharmacological inhibition of Notch activity during homeostasis decreases *fgf3* expression it has no effect on Fgf target genes (*Figure 8L–P'*). Additionally, activating Notch signaling by over-expression of a Notch intracellular domain inhibits proliferation and

differentiation during regeneration in both wild type and *fgf3*[-/-] (*Figure 8R,T,V*). This finding shows that Notch inhibits Wnt signaling even in the absence of Fgf3. These data argue that Fgf and Notch signaling are functioning largely in parallel. We therefore postulate that the transient down regulation of both of these pathways immediately after hair cell death is required to maximally activate Wnt signaling and induce proliferation. Nevertheless, the Fgf and Notch pathways are reactivated before hair cell regeneration is complete (*Jiang et al., 2014*) and play additional roles later in regeneration, such as ensuring that not too many support cells differentiate. As such, in mammals a short inhibition of one or both of these pathways is more likely to result in functional regeneration than prolonged treatments.

## Conclusion

The scRNA-Seq analysis revealed previously unidentified support cell populations and combined with in situ validation of these cell clusters identified lineages that either lead to stem cell self-renewal or hair cell differentiation. Importantly, we have identified the cascade of gene activation and repression that leads to hair cell differentiation. Our analyses led to the hypothesis that some of the support cell populations are involved in signaling to trigger regeneration, which we tested by scRNA-Seq analyses of *fgf3* mutants that strikingly show increased proliferation and hair cell regeneration. These experiments identified *fgf3* targets that we could not identify in bulk RNA-Seq analyses. Having characterized the support cell transcriptome of a regenerating species allows us to identify commonalities and differences with mouse support cells that do not trigger a meaningful regenerative response (*Burns et al., 2015*; *Maass et al., 2016*). Such a comparison will become even more powerful once adult mouse single cell transcriptomes of support cells are available.

# Materials and methods

**Key resources table**

| Reagent type (species) or resource | Designation | Source or reference | Identifiers | Additional information |
|---|---|---|---|---|
| Genetic reagent (*D. rerio*) | ET(krt4:EGFP) SqGw57a | (*Kondrychyn et al., 2011*) | SqGw57a | |
| Genetic reagent (*D. rerio*) | Tg(pou4f3:GAP-GFP)[s356t] | (*Xiao et al., 2005*) | s356t; RRID:ZFIN_ZDB-GENO-100820-2 | |
| Genetic reagent (*D. rerio*) | fgf3[t26212] | (*Herzog et al., 2004*) | t26212; RRID:ZFIN_ZDB-ALT-040716-16 | |
| Genetic reagent (*D. rerio*) | Tg(hsp70l:dkk11 b-GFP)[w32tg] | (*Stoick-Cooper et al., 2007*) | w32tg; RRID:ZFIN_ZDB-GENO-100420-26 | |
| Genetic reagent (*D. rerio*) | Tg(hsp70l:MYC-notch1a, cryaa:Cerulean)[fb12Tg] | (*Zhao et al., 2014*) | fb12Tg; RRID:ZFIN_ZDB-ALT-140522-5 | |
| Genetic reagent (*D. rerio*) | Tg(hsp70l:dnfgfr 1a-EGFP)[pd1tg] | (*Lee et al., 2005*) | pd1tg; RRID:ZFIN_ZDB-ALT-060322-2 | |
| Genetic reagent (*D. rerio*) | TgBAC(prox1a:KALTA4, 4xUAS-ADV.E1b:TagRFP)[nim5Tg] | (*van Impel et al., 2014*) | nim5Tg; RRID:ZFIN_ZDB-ALT-140521-3 | |
| Genetic reagent (*D. rerio*) | Tg(EPV.Tp1-Mm u.Hbb:EGFP)[um14] | (*Parsons et al., 2009*) | um14; RRID:ZFIN_ZDB-GENO-090626-1 | |
| Genetic reagent (*D. rerio*) | Et(krt4:EGFP)[sqet4ET] | (*Parinov et al., 2004*) | sqet4ET; RRID:ZFIN_ZDB-GENO-070702-7 | |
| Genetic reagent (*D. rerio*) | Et(krt4:EGFP)[sqet20ET] | (*Parinov et al., 2004*) | sqet20ET; RRID:ZFIN_ZDB-ALT-070628-20 | |

*Continued on next page*

*Continued*

| Reagent type (species) or resource | Designation | Source or reference | Identifiers | Additional information |
|---|---|---|---|---|
| Genetic reagent (*D. rerio*) | *Tg (fgf3:H2B-mturquoise2)$^{psi60Tg}$* | This paper | psi60Tg | |
| Genetic reagent (*D. rerio*) | *fgfr1a$^{sa38715}$* | This paper | sa38715; RRID:ZFIN_ZDB-ALT-161003-16150 | |
| Genetic reagent (*D. rerio*) | *fgfr2$^{sa30975}$* | This paper | sa30975; RRID: ZFIN_ZDB-ALT-160601-753 | |
| Antibody | Mouse monoclonal Anti-BrdU | Sigma Aldritch | 1170376001; RRID:AB_2313622 | IHC (1/200) |
| Antibody | Rabbit poly clonal Anti-GFP | Thermo Fisher Scientific | A11122; RRID: AB_10073917 | IHC (1/400) |
| Antibody | Alexa Fluor 568 Goat anti-mouse monoclonal | Thermo Fisher Scientific | A11004; RRID:AB_141371 | IHC (1/1000) |
| Antibody | Alexa Fluor 488 Goat anti-rabbit polyclonal | Thermo Fisher Scientific | A11034; RRID:AB_2576217 | IHC (1/1000) |
| Sequence based reagent | crRNA | IDT | GGCCATGGAAACTAAATCTG | |
| Peptide, r ecombinant protein | Cas9 | PNA Bio | CP01 | 1 μM |
| Commercial assay or kit | Chromium Single Cell 3' Library and Gel Bead Kit | 10X Genomics | 120267 | |
| Commercial assay or kit | SMART-Seq v4 Ultra Low Input RNA kit | Takara | 634888 | |
| Chemical compound, drug | BrdU | Sigma Aldritch | B9285 | 10 mM |
| Chemical compound, drug | EdU | Carbosynth | NE08701 | 3.3 mM |
| Chemical compound, drug | Neomycin sulfate | Sigma Aldritch | N6386 | 300 μM |
| Chemical compound, drug | Alexa Fluor-594 Azide | Thermo Fisher Scientific | A10270 | 2.5 μM |
| Chemical compound, drug | LY411575 | Selleckchem | S2714 | 50 μM |
| Software, algorithm | Cell Ranger (v1.3.1 WT data set; v2.1.1 for *fgf3 data sets*) | 10X Genomics | | |
| Software, algorithm | Seurat (v2.3.4) | (**Butler et al., 2018**) | | |
| Software, algorithm | Scanpy (v1.3.2) | (**Wolf et al., 2018a**) | | |

## Fish lines and husbandry

ET(krt4:EGFP)SqGw57a (**Kondrychyn et al., 2011**), Tg(pou4f3:GAP-GFP)$^{s356t}$ (**Xiao et al., 2005**), fgf3$^{t26212}$ (**Herzog et al., 2004**), Tg(hsp70l:dkk11b-GFP)$^{w32tg}$ (**Stoick-Cooper et al., 2007**), Tg (hsp70l:MYC-notch1a,cryaa:Cerulean)$^{fb12Tg}$ (**Zhao et al., 2014**), Tg(hsp70l:dnfgfr1a-EGFP)$^{pd1tg}$ (**Lee et al., 2005**), TgBAC(prox1a:KALTA4,4xUAS-ADV.E1b:TagRFP)$^{nim5Tg}$ (**van Impel et al., 2014**), Tg(EPV.Tp1-Mmu.Hbb:EGFP)$^{um14}$ (**Parsons et al., 2009**), Et(krt4:EGFP)$^{sqet4ET}$ and Et(krt4:EGFP)$^{sqet20ET}$ (**Parinov et al., 2004**). Tg(6xNFkB:EGFP)$^{nc1}$ (**Kanther et al., 2011**). fgfr1a$^{sa38715}$ and fgfr2$^{sa30975}$ are from the Sanger Institute Zebrafish Mutation project. All experiments were performed according to guidelines established by the Stowers Institute IACUC review board.

## Generation of *Tg(fgf3:H2B-mturquoise2)$^{psi60Tg}$*

H2B-mturquoise2 was placed near the 5' region of *fgf3* using non-homologous repair with CRISPR/Cas9 (*Auer et al., 2014*; *Kimura et al., 2014*). A CRISPR recognition site (GGCCATGGAAACTAAATCTGCGG) was chosen 584 bp in front of the *fgf3* transcription start site using CRISPRscan (*Moreno-Mateos et al., 2015*). The same recognition site was cloned by PCR onto both ends of a construct containing a 56 bp β-actin minimal promoter driving human-H2B fused at the c-terminus with mturquoise2 followed by the SV40 polyA from the Tol2 kit (*Kwan et al., 2007*). This plasmid was mixed with the crRNA (GGCCATGGAAACTAAATCTG, IDT), tracrRNA (IDT) and Cas9 protein (PNA Bio) and complexed for 10 min at room temperature then placed on ice. The complex was then injected into the cell of a one cell stage zebrafish embryos from a cross of *fgf3$^{t26212}$* to wild type. Integrated DNA will just contain the minimal β-actin promoter, H2B-mturquoise2 and the polyA sequence without any plasmid DNA. In a few larvae, fluorescence could be seen by 24 hrs and onward. Fluorescent embryos were sorted and raised to identify founders that showed H2B-mturquoise expression similar to *fgf3* expression. The founder generated also has the *fgf3$^{t26212}$* allele and is therefore heterozygous for the mutation. These fish are viable and fertile and show no obivious phenotypes.

## Sensory hair cell ablation

For hair cell ablation experiments 5 days post fertilization (dpf) fish were exposed to 300 µM Neomycin (Sigma-Aldrich) for 30 min at room temperature. Neomycin was then washed out and larvae were allowed to recover at 28°C for as long as the experiment lasted.

## Proliferation analysis

BrdU (Sigma-Aldrich) was added at 10 mM with 1% DMSO in E2. Larvae were treated for 24 hrs then fixed in 4% paraformaldehyde overnight at 4°C. Mouse anti-BrdU (Sigma-Aldrich) and rabbit anti-GFP (Invitrogen) immunohistochemistry with DAPI (Invitrogen) counterstain was performed as described (*Lush and Piotrowski, 2014b*). EdU (Carbosynth) was added at 3.3 mM with 1% DMSO in E2. Larvae were treated for 24 hrs then fixed in 4% paraformaldehyde overnight at 4°C. For staining, larvae were washed 3 times 10 min each in PBS/0.8% Trition-X (PBSTX), blocked for 1 hr in 3%BSA/PBSTX, washed 3 times 5 min each in PBS then put in fresh staining solution for 30 min. Staining solution contains 1xTris buffered saline, 0.1% Tween-20, 2 mM CuSO$_4$, 2.5 mM Alexa-594-Azide (Invitrogen) and 50 mM ascorbic acid. After staining, larvae were washed extensively in PBSTX. Larvae were then processed for anti-GFP immunohistochemistry and DAPI staining as described above. Stained larvae were imaged on a Zeiss LSM780 confocal microscope at 40X. Three posterior lateral line neuromasts (L1, L2 and L3) were imaged per fish. Cell numbers were manually counted in Imaris software. Spatial positioning was performed as described (*Romero-Carvajal et al., 2015*; *Venero Galanternik et al., 2016*).

## In situ hybridization and alkaline phosphatase staining

In situ hybridization was performed as described with modifications (*Kopinke et al., 2006*) and http://research.stowers.org/piotrowskilab/in-situ-protocol.html. Incubation time in proteinase K (Roche) depended on the batch, and varied from 2 to 5 min at room temperature. Prehybridization was performed for at least 2 hrs at 65°C. Probes used were *fgfr1a*, *fgf3*, *fgf10a*, and *etv4* (*Aman and Piotrowski, 2009*), *fgfr2* (*Rohs et al., 2013*), *dld* (*Jiang et al., 2014*), *wnt2* (*Poulain and Ober, 2011*), *wnt10a* (*Lush and Piotrowski, 2014b*), *wnt11r* (*Duncan et al., 2015*), *gfp* (*Dorsky et al., 2002*), *atoh1a* and *notch3* (*Itoh and Chitnis, 2001*) and *sfrp1a* (*Tendeng and Houart, 2006*) Additional probes were generated by PCR from zebrafish cDNA and cloned into the topo-pCRII vector (Invitrogen). See Data file 16 for list or primers used. Alkaline phosphatase staining was performed as described (*Lush and Piotrowski, 2014a*).

## Heat shock paradigm

Heat shock induction of transgene expression varied depending on the transgenic line used. Initially 5dpf larvae were heat shocked for 1 hr at 37°C then put back at 28°C for 1 hr. Larvae were then heat shocked for 1 hr at a higher temperature (39°C for *notch1a*-intraceullar and 40°C for *dkk1a* and *dnfgfr1a*), put at 28°C for 1 hr, followed by another higher temperature heat shock (39°C or 40°C) for

1 hr. Larvae were then allowed to recover for 1 hr at 28°C then fixed in 4% paraformaldehyde for in situ hybridization or continued with neomycin and proliferation analysis. For neomycin experiments, larvae were treated for 30 min in 300 μM, then extensively washed and immediately placed in EdU as described above. Larvae were then heat shock at the higher temperature followed by 2 hrs recovery at 28°C. The 1 hr heat shock followed by 2 hrs recovery was repeated for a total of 24 hrs post neomycin treatment. Larvae were then fixed in 4% paraformaldehyde overnight at 4°C.

## Time-lapse imaging and confocal imaging

*TgBAC(prox1a:KALTA4,4xUAS-ADV.E1b:TagRFP)* were crossed to *Tg(pou4f3:GAP-GFP)$^{s356t}$* and larvae were raised to 5dpf. Time-lapse imaging after neomycin treatment was carried out on a Zeiss LSM780 confocal microscope as described (*Venero Galanternik et al., 2016*). *Tg(fgf3:H2B-mturquoise2)$^{psi60Tg}$/Tg(pou4f3:GAP-GFP)$^{s356t}$* or *Tg(fgf3:H2B-mturquoise2)$^{psi60Tg}$/TgBAC(prox1a:KALTA4,4xUAS-ADV.E1b:TagRFP)* double transgenic larvae were imaged live as above. Three-dimensional rendering and image analyses were done using Imaris (Bitplane).

## Sample preparation for scRNA-seq

### Embryo dissociation and FACS

5 dpf *Tg(pou4f3:GAP-GF)/Tg(GW57a:GFP)* zebrafish embryos were dissociated by adding 1.5 ml (per 100 embryos) of 0.25% Trypsin-EDTA (Thermo Fisher Scientific, Waltham, MA. USA) and triturated with 1 ml pipette tip for 3 min on ice. To collect dissociated cells, cells were filtered with 70 μm Cup Filcons (BD Biosciences, San Jose, CA. USA) and washed with ice-cold DPBS (centrifugation at 2000 rpm for 6 mins at 4°C). Cells were stained with Hoechst 33342 (final concentration: 0.005 μg/μl) or Draq5 (1:2000) (biostatus, UK) and 7-AAD (final concentration: 0.5 μg/ml) or DAPI (5 μg per ml) to gate out dead cell populations. FACS was performed at the Cytometry Core facility (Stowers Institute for Medical Research) using BD Influx Cell Sorter (BD Biosciences, San Jose, CA. USA).

### 10X Chromium scRNA-seq library construction

scRNA-seq was carried out with 10X Chromium single cell platform (10X Genomics, Pleasanton, CA. USA). FAC-sorted live or MeOH-fixed cells were used as the input source for the scRNA-seq. MeOH-fixed cells were rehydrated with rehydration buffer (0.5% BSA and 0.5 U/μl RNase-inhibitor in ice-cold DPBS) following manufacturer's instructions (10X Genomics). The maximum recommended volume of single cell suspension (34 μl) was loaded on a Chromium Single Cell Controller (10x Genomics) targeting ~1500–2000 cells per sample. Chromium Single Cell 3' Library and Gel Bead Kit v2 (10X Genomics) was used for libraries preparation according to manufacturer's instructions. Resulting short fragment libraries were checked for quality and quantity using an Agilent 2100 Bioanalyzer and Invitrogen Qubit Fluorometer. Libraries were sequenced to a depth of ~160–330M reads each on an Illumina HiSeq 2500 instrument using Rapid SBS v2 chemistry with the following paired read lengths: 26 bp Read1, 8 bp I7 Index and 98 bp Read2.

### scRNA-Seq read alignment and quantification

Raw reads were demultiplexed and aligned to version 10 of the zebrafish genome (GRCz10) using the Cell Ranger pipeline from 10X Genomics (version 1.3.1 for wild type and version 2.1.1 for *fgf3$^{-/-}$* data sets). 1,666 cell barcodes were obtained for wild type embryos, 1932 for *fgf3* siblings and 1459 for *fgf3* mutants. These quantities were estimated using Cell Ranger's barcode ranking algorithm, which estimates cell counts by obtaining barcodes that vary within one order of magnitude of the top 1 percent of barcodes by top UMI counts. The resulting barcodes (henceforth referred to as cells) were used to generate a UMI count matrix for downstream analyses. Data deposition: the BAM files and count matrices produced by Cell Ranger have been deposited in the Gene Expression Omnibus (GEO) database, www.ncbi.nlm.nih.gov/geo (accession no. GSE123241).

## Quality control, dimensional reduction, and cell classification

Subsequent analyses on UMI count matrix for all three data sets were performed using the R package Seurat (version 2.3.4, *Butler et al., 2018*) following the standard workflow outlined in the pbmc3k example on the Satija lab webpage (https://satijalab.org). Both *fgf3* sibling and mutant data sets were analyzed independently using the same parameters and arguments, and then each data

set was merged using the Seurat function MergeSeurat(). Initial gene quality control was performed by filtering out genes expressed in less than 3 cells for the WT data set, and five for the *fgf3* sibling/ mutant data sets. The remaining UMI counts were then log normalized. Gene selection for dimensional reduction was accomplished using the Seurat function FindVariableGenes() with the following arguments for the WT data set: x.low.cutoff = 0.001, x.high.cutoff = 3.0, and y.cutoff = 0.5; and for the fgf3 sibling/mutant data sets: x.low.cutoff = 0.20 and y.cutoff = −0.20. Following gene selection, all log-normalized expression values were then scaled and centered using ScaleData(). For dimensional reduction, we chose to use the first six principal components (PCs) for wild type and the first 19 for the *fgf3* sibling/mutant data sets. PCs were chosen according to the PCA elbow plot, which orders PCs from highest to lowest based on the percentage of variance explained by each PC. Thus, each set of PCs chosen showed the highest percentage of variance explained on the elbow plot. Next, we performed clustering on each set of principal components, and for two-dimensional visualization, we performed a second round of dimensional reduction using t-SNE. The dendrogram in *Figure 3A* was generated using the Seurat function BuildClusterTree(). This function performs hierarchical clustering based on a Euclidian distance matrix, which is created from the average expression of the variable genes (the same set used for dimensional reduction) in each of the 14 clusters. Cells for all data sets were classified according to their marker gene expression. Markers for each cell type were identified based on their differential gene expression using the Seurat function FindAllMarkers(). Genes with an adjusted p-value less than 0.001 were retained. For the pairwise comparison between *fgf3* mutant and sibling data sets, we used the function FindMarkers() and retained genes with a fold change greater than 0.10, or less than −0.10. Genes differentially expressed between dendrogram nodes were calculated using the function FindAllMarkersNode(), and we kept the top 100 genes with the highest p-values for each node comparison. All three data sets contained non-specific cell types that contained markers for skin and blood cells which were removed from the final analysis.

## Pseudotime analysis

All data contained within our processed Seurat object for the wild type data set was converted to the AnnaData format for pseudotime analysis in Scanpy (version 1.2.2, *Wolf et al., 2018a*), using the Seurat function convert. We recalculated *k*-nearest neighbors at k = 15 and chose cluster 14 as our putative 'stem cell' population. Pseudotime was calculated using Scanpy's partitioned-based graph abstraction function, paga. To visualize gene expression in pseudotime, cells from clusters 14, 4, 2, and one were subsetted into a separate matrix and ordered according to their pseudotime values from least to greatest. Next, we chose genes differentially expressed between clusters, and differentially expressed between selected nodes in our cluster dendrogram. The resulting genes were ordered according to our understanding of neuromast biology and pseudotime expression patterns. The final count matrix was then log normalized and rendered as a heatmap using the python package Seaborn (version 0.8.1).

## GO term analysis

The GO term analysis was performed in DAVID (Database for Annotation, Visualization and Integrated Discovery, (*Huang et al., 2009*).

## Web apps and data repository

The web app (https://piotrowskilab.shinyapps.io/neuromast_homeostasis_scrnaseq_2018/) for our homeostatic data set was generated using the Shiny framework in R. Shiny allows one to create a web-based GUI (graphical user interface) for data exploration using R code (the source code for our app can be found at https://github.com/Piotrowski-Lab/Shiny-Apps; *Diaz, 2019*; copy archived at https://github.com/elifesciences-publications/Shiny-Apps). The user can choose a zebrafish gene(s) of interest, provided it is expressed in our dataset, and view expression values on a heatmap, violin, or t-SNE plot. Instructions are provided in the welcome page of the web app. We are also in the process of uploading the data into gene Expression Analysis Resource (gEAR), a website for visualization and comparative analysis of multi-omic data, with an emphasis on hearing research (https://umgear.org).

## RT-qPCR

### RNA extraction and cDNA synthesis

Total RNA was extracted from FAC-sorted cells using Trizol (Thermo Fisher Scientific, Waltham, MA. USA), chloroform and isopropanol. During isopropanol precipitation, to enhance the pellet visibility, GlycoBlue coprecipitant (Thermo Fisher Scientific, Waltham, MA. USA) was used. Following that, total RNA was washed with 80% ice-cold Ethanol and resuspended in RNase-free water. The first-strand cDNA synthesis and cDNA amplification were done using SMART-Seq v4 Ultra Low Input RNA kit (Takara Bio USA, Mountain View, CA. USA) following manufacturer's instructions.

### Quantitative PCR

Q-PCR was carried out using ABI SYBR Green master mix (Applied Biosystems, Foster City, USA) in the QuantStudio 7 Real-Time PCR System with a 384-Well Block (Applied Biosystems, Foster City, USA). The reaction program consisted of four steps: UDG treatment (50°C for 10 min), quantitation (40 cycles of 95°C for 15 s, 60°C for 60 s) and melting curve analysis (95°C for 15 s, 60°C for 60 s, 95°C for 15 s). The experiment was performed in triplicate. All the signals were normalised to the $ef1\alpha$ expression level. All primer sequences are provided in Data file 15.

## Acknowledgments

We are grateful to Drs. A Sánchez Alvarado and J Sandler for valuable comments on the manuscript, and C Burns, S McFarlane, E Ober, J Rawls, A Nechiporuk, V Korsh, G Crump, K Poss, M Parrish, and B Appel for strains and reagents. We are also thankful to members of the Piotrowski lab for insightful discussions. We thank the Stowers Institute Aquatics Facility for excellent fish care and M Miller for help with scientific illustration. This work was funded by an NIH (NIDCD) award 1R01DC015488-01A1 to TP, the Hearing Health Foundation, and by institutional support from the Stowers Institute for Medical Research.

## Additional information

### Funding

| Funder | Grant reference number | Author |
|---|---|---|
| National Institute on Deafness and Other Communication Disorders | 1R01DC015488-01A1 | Tatjana Piotrowski |
| Hearing Health Foundation | | Tatjana Piotrowski |

The funders had no role in study design, data collection and interpretation, or the decision to submit the work for publication.

### Author contributions

Mark E Lush, Conceptualization, Data curation, Formal analysis, Investigation, Visualization, Methodology, Writing—original draft; Daniel C Diaz, Conceptualization, Data curation, Software, Formal analysis, Validation, Investigation, Visualization, Writing—original draft; Nina Koenecke, Conceptualization, Software, Methodology, Writing—review and editing; Sungmin Baek, Formal analysis, Validation, Investigation, Methodology, Writing—review and editing; Helena Boldt, Validation, Methodology; Madeleine K St Peter, Visualization, Methodology; Tatiana Gaitan-Escudero, Validation, Visualization, Methodology; Andres Romero-Carvajal, Formal analysis, Methodology; Elisabeth M Busch-Nentwich, Resources; Anoja G Perera, Kathryn E Hall, Allison Peak, Jeffrey S Haug, Methodology; Tatjana Piotrowski, Conceptualization, Data curation, Formal analysis, Supervision, Funding acquisition, Investigation, Visualization, Writing—original draft, Project administration

### Author ORCIDs

Mark E Lush ⬤ http://orcid.org/0000-0002-2128-524X
Daniel C Diaz ⬤ http://orcid.org/0000-0002-5582-4686

Andres Romero-Carvajal (iD) http://orcid.org/0000-0002-2570-1749
Elisabeth M Busch-Nentwich (iD) http://orcid.org/0000-0001-6450-744X
Tatjana Piotrowski (iD) http://orcid.org/0000-0001-8098-2574

## Ethics

Animal experimentation: This study was performed in strict accordance with the recommendations in the Guide for the Care and Use of Laboratory Animals of the National Institutes of Health. All of the animals were handled according to approved institutional animal care use committee (IACUC) protocol (#2017-0176) of the Stowers Institute for Medical Research.

## Decision letter and Author response

Decision letter https://doi.org/10.7554/eLife.44431.048
Author response https://doi.org/10.7554/eLife.44431.049

## Additional files

### Supplementary files

• Supplementary file 1. Related to *Figure 1*: excel file of genes that are expressed in at least three cells.
DOI: https://doi.org/10.7554/eLife.44431.022

• Supplementary file 2. Related to *Figure 1E*: excel file of cluster marker genes.
DOI: https://doi.org/10.7554/eLife.44431.023

• Supplementary file 3. Related to *Figure 1E*: t-SNE plots of all cluster marker genes.
DOI: https://doi.org/10.7554/eLife.44431.024

• Supplementary file 4. Related to *Figure 2D*: excel file of cell cycle genes.
DOI: https://doi.org/10.7554/eLife.44431.025

• Supplementary file 5. Related to *Figure 2D*: t-SNE plots of cell cycle genes.
DOI: https://doi.org/10.7554/eLife.44431.026

• Supplementary file 6. Related to *Figure 2—figure supplement 2*: excel file of zebrafish orthologs of human deafness genes.
DOI: https://doi.org/10.7554/eLife.44431.027

• Supplementary file 7. Related to *Figure 3A*: excel files of differentially expressed genes between nodes (dendrogram).
DOI: https://doi.org/10.7554/eLife.44431.028

• Supplementary file 8. Related to *Figure 3A*: heatmaps of dendrogram node genes.
DOI: https://doi.org/10.7554/eLife.44431.029

• Supplementary file 9. Related to *Figure 4A–H*: excel file of hair cell lineage genes.
DOI: https://doi.org/10.7554/eLife.44431.030

• Supplementary file 10. Related to *Figure 4A–H*: t-SNE plots of hair cell lineage genes.
DOI: https://doi.org/10.7554/eLife.44431.031

• Supplementary file 11. Related to *Figure 4I*: excel file of hair cell genes ordered along pseudotime.
DOI: https://doi.org/10.7554/eLife.44431.032

• Supplementary file 12. Related to *Figure 4—figure supplement 1*: excel file of cilia genes.
DOI: https://doi.org/10.7554/eLife.44431.033

• Supplementary file 13. Related to *Figure 7*: excel file of cluster markers in *fgf3*$^{-/-}$ scRNA-Seq.
DOI: https://doi.org/10.7554/eLife.44431.034

• Supplementary file 14. Related to *Figure 7C*, S9B: differentially expressed genes in the *fgf3*$^{-/-}$.
DOI: https://doi.org/10.7554/eLife.44431.035

• Supplementary file 15. Primer table for RT-qPCR.
DOI: https://doi.org/10.7554/eLife.44431.036

• Supplementary file 16. Related to *Figures 1*, *2*, *5*, *7* and *8*: primers used for in situ probes.
DOI: https://doi.org/10.7554/eLife.44431.037

• Transparent reporting form
DOI: https://doi.org/10.7554/eLife.44431.038

## Data availability

BAM files and count matrices produced by Cell Ranger have been deposited in the Gene Expression Omnibus (GEO) database, www.ncbi.nlm.nih.gov/geo (accession no. GSE123241).The original data underlying this manuscript can be accessed from the Stowers Original Data Repository at http://www.stowers.org/research/publications/libpb-1382.

The following datasets were generated:

| Author(s) | Year | Dataset title | Dataset URL | Database and Identifier |
|---|---|---|---|---|
| Diaz DC | 2018 | Single cell RNA-Seq reveals Fgf signaling dynamics during sensory hair cell regeneration | https://www.ncbi.nlm.nih.gov/geo/query/acc.cgi?acc= GSM3498552 | NCBI Gene Expression Omnibus, GSM3498552 |
| Diaz DC, Baek S, Boldt H, Perera A, Hall K, Peak A, Haug JS, Piotrowski T | 2018 | Single cell RNA-Seq reveals Fgf signaling dynamics during sensory hair cell regeneration | https://www.ncbi.nlm.nih.gov/geo/query/acc.cgi?acc=GSM3498553 | NCBI Gene Expression Omnibus, GSM3498553 |
| Diaz DC, Baek S, Boldt H | 2018 | Single cell RNA-Seq reveals Fgf signaling dynamics during sensory hair cell regeneration | https://www.ncbi.nlm.nih.gov/geo/query/acc.cgi?acc=GSM3498554 | NCBI Gene Expression Omnibus, GSM3498554 |
| Lush ME, Diaz DC, Koenecke N, Baek S, Boldt H, St Peter MK, Gaitan-Escudero T, Romero-Carvajal A, Busch-Nentwich EM, Perera AG, Hall KE, Peak A, Haug JS, Piotrowski T | 2019 | scRNA-Seq reveals distinct stem cell populations that drive hair cell regeneration after loss of Fgf and Notch signaling | https://www.stowers.org/research/publications/libpb-1382 | Stowers Original Data Repository, libpb-1382 |

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
