## [Decision Letter]

Thank you for submitting your article "scRNA-Seq reveals distinct stem cell populations that drive hair cell regeneration after loss of FGF and Notch signaling" for consideration by *eLife*. Your article has been reviewed by two peer reviewers, and the evaluation has been overseen by Tanya Whitfield as the Reviewing Editor and Marianne Bronner as the Senior Editor. The following individual involved in review of your submission has agreed to reveal his identity: Shawn Burgess (Reviewer #2).

As you will see, both reviewers are very enthusiastic about the study, and only have very minor suggestions for revision, where small changes would help to clarify the data, conclusions or methodology. Please attend to these where possible. The full reviews are appended below for your information.

Reviewer #1:

The zebrafish lateral line neuromast has emerged as a powerful model to study hair cell regeneration. Although the behavior of supporting cells and their neighboring mantle cells during homeostatic hair cell turnover and after hair cell killing has been well-characterized, the genetic pathways underlying these responses has been poorly understood. The present study is a beautiful single cell RNA-seq analysis of the different cell populations, and provides additional new insights into the signaling and genetic pathways underlying regeneration in neuromasts. The results will help to inform studies of hair cell regeneration failure in mammals and will also likely shed light on regenerative behavior in the inner ear of birds.

The manuscript is beautifully illustrated and the results are – with one or two exceptions I describe below – extremely compelling. Most of my comments are not serious, and I do not feel that significant new experiments are required to revise this study.

Reviewer #1 (Minor Comments):

A number of places in the text would benefit from more explanation or clarification:

1) The last part of the first paragraph of the Results makes a distinction between priml and primll neuromasts and states that "gene expression patterns" are only discussed in priml-derived neuromasts. I believe the authors are referring to in situ hybridization and reporter gene expression in the figures, not the sc-RNA seq data. This should be made clearer (and perhaps just moved to the Materials and methods section).

2) The second paragraph of the Results describes how marker gene expression is used to assign different clusters to different cell types. It would be helpful to give examples of the genes that were used to make these assignments. I would also recommend being consistent with the use of "group" versus "cluster" throughout.

3) Figure 2E – I would add labels to identify the red and green cells in the panel, not just in the legend.

4) Subsection “Heatmaps and the ordering of cells along pseudotime reveals how gene networks change in different lineages” – how was the dendrogram in Figure 3A generated?

5) In introducing Figure 3A and B, the authors should remind the non-specialist reader of the previous work that has identified different patterns of supporting cell behavior (amplifying, differentiating, etc.) in the neuromast.

6) In the PAGA analysis, the authors should spend a little more time describing the how a statistical measure of the degree of connectivity relates to the direction of differentiation for the benefit of non-specialists.

I have a number of suggestions on the Wnt/FGF part of the paper:

7) – "Also, the Wnt target gene *wnt10a* is upregulated, illustrating that Wnt signaling is increased in *fgf3* mutants (Figure 7G and G')". Is there more evidence from other transcripts in the *fgf3* mutant RNA-seq data in support of this increase in Wnt signaling?

8) On a similar theme, – "FGF signaling does not act upstream of Notch signaling, as Notch pathway members are not affected by in situ hybridization in *fgf3*^-/-^". A small number of Notch pathway genes are offered in support of this negative result, and only by in situ, not RNA-seq. For both Notch and Wnt signaling, some kind of gene ontology analysis or just a list of Notch and Wnt pathway genes would be good. I am aware that limited coverage of the 10x Genomics platform precludes an exhaustive catalog of signaling pathways, but additional data would strengthen both claims.

9) – "*fgf3* on the other hand, is slightly downregulated after Notch signaling inhibition with a γ-secretase inhibitor". The data supporting this point seems quite weak. My suggestion is that the authors either remove these data and point from the paper, or else substantiate it better – for example, by treating their neuromast reporter line with γ secretase inhibitors, sorting the cells and measuring *fgf3* message by qRT-PCR. It is not an essential point of the paper, so I would be fine with just removing it.

10) I recommend Supplementary Figure 10 be included in the main figures, not as a supplementary figure. As mentioned above, I would also be fine with removing the weak dotted line from Notch to *fgf3*.

A few points on the Discussion:

11). I was not very convinced by the attempts of the authors to bolster the references to stem cells in other systems. For example, the shared expression of *tspan1* and its planarian pseudo-homologue is a bit weak. Similarly, in the subsection “Support cells share gene expression profiles with stem cells in other organs”, although GLAST and FABP7 are SVZ markers, they're also expressed in mature astrocytes and some mature mammalian supporting cells. Where the authors state of neuromast supporting cells that "their gene expression profiles share many genes with stem cells in the CNS" – how many is "many"? And what genes are these in addition to the five listed in that paragraph?

12) Subsection “Signals that control multipotency versus differentiation”. Care should be taken in interpretation of negative expression of Notch reporters as being indicative of a lack of Notch signaling, as these reporters can be extremely context dependent, and a negative result with one measurement is potentially problematic. Moreover, work in the mammalian organ of Corti has shown that some supporting cells (inner pillar cells) can have both Notch and FGF signaling active in them, and their conversion to hair cells requires both pathways to be blocked.

13) Subsection “*Fgf3*

inhibits Wnt signaling and proliferation possibly via Sost” refers to the role of Wnt signaling in the mammalian organ of Corti. The authors should reference the recent Development paper from the Edge and Dabdoub labs. Also, there are several databases of supporting cell gene expression that the authors could use to look at sclerostin expression in mammals.

14) Finally, I would recommend deleting the section titled "A role for FGF is (sic) specification?" The points made here are not especially strong and don't contribute significantly to the main messages of the paper.

Some additional very minor points:

15) In the Abstract, I suggest changing "we implemented a shiny application" to "we implemented a web-based application". It is likely that many people will be ignorant of the Shiny platform and may think this is a typo! Similarly, the reference to the "shiny app" would benefit from a web link.

16) In Figure 1B, I recommend labeling the supporting cells in addition to the hair and mantle cells.

Reviewer #2:

The authors have submitted a scRNA-seq analysis of lateral line neuromasts in zebrafish 5dpf larvae. ≈1200 cells were purified via FAC sorting based on transgene expression in the hair cells and support cells and profiled using the 10X Genomics Chromium platform. Typical analyses were performed using various clustering algorithms, and 7 distinct cell types in 14 clusters were identified. The authors have generated a useful website where anyone can query the data. From the initial analyses, hypotheses regarding the role of *fgf3* in regulating regeneration and its relationship to notch and wnt signaling were tested.

Frankly, I have no major concerns with the data as presented. The work is carefully done, uses generally accepted approaches and analyses, and would be of broad interest to the scientific community.

Reviewer #2 (Minor Comments):

Two minor comments:

- In figure 1K, *isl1* is capitalized, should be lower case.

- The act of dissociating zebrafish embryos into single cells is likely to trigger a global injury response. Some discussion of false positive expression of these genes should be discussed.

---

## [Author Response]

Reviewer #1 (Minor Comments):A number of places in the text would benefit from more explanation or clarification:1) The last part of the first paragraph of the Results makes a distinction between priml and primll neuromasts and states that "gene expression patterns" are only discussed in priml-derived neuromasts. I believe the authors are referring to in situ hybridization and reporter gene expression in the figures, not the sc-RNA seq data. This should be made clearer (and perhaps just moved to the Materials and methods section).

Thank you for the comment. We have clarified the relevant text.

2) The second paragraph of the Results describes how marker gene expression is used to assign different clusters to different cell types. It would be helpful to give examples of the genes that were used to make these assignments. I would also recommend being consistent with the use of "group" versus "cluster" throughout.

We now reference Figure 1E (list of cluster marker genes) and have substituted the term ‘group’ with ‘cluster’.

3) Figure 2E – I would add labels to identify the red and green cells in the panel, not just in the legend.

We apologize if we misinterpret the suggestion but the cells are labeled in the current version.

4) Subsection “Heatmaps and the ordering of cells along pseudotime reveals how gene networks change in different lineages” – how was the dendrogram in Figure 3A generated?

We have now added a description to the Materials and methods.

5) In introducing Figure 3A and B, the authors should remind the non-specialist reader of the previous work that has identified different patterns of supporting cell behavior (amplifying, differentiating, etc.) in the neuromast.

We added a sentence in the first paragraph of the subsection “Heatmaps and the ordering of cells along pseudotime reveals how gene networks change in different lineages” reminding readers of which lineages were previously identified in time-lapse analyses.

6) In the PAGA analysis, the authors should spend a little more time describing how a statistical measure of the degree of connectivity relates to the direction of differentiation for the benefit of non-specialists.

We rewrote this section and added a better description (subsection “Heatmaps and the ordering of cells along pseudotime reveals how gene networks change in different lineages”).

I have a number of suggestions on the Wnt/FGF part of the paper:7) "Also, the Wnt target gene wnt10a is upregulated, illustrating that Wnt signaling is increased in fgf3 mutants (Figure 7G and G')". Is there more evidence from other transcripts in the fgf3 mutant RNA-seq data in support of this increase in Wnt signaling?

Unfortunately, the scRNASeq data did not reveal other significantly upregulated Wnt pathway targets. Even *wnt10a* was not identified in the scRNASeq data, even though it is clearly upregulated by in situ hybridization and qRTPCR.

8) On a similar theme, "FGF signaling does not act upstream of Notch signaling, as Notch pathway members are not affected by in situ hybridization in fgf3^-/-^". A small number of Notch pathway genes are offered in support of this negative result, and only by in situ, not RNA-seq. For both Notch and Wnt signaling, some kind of gene ontology analysis or just a list of Notch and Wnt pathway genes would be good. I am aware that limited coverage of the 10x Genomics platform precludes an exhaustive catalog of signaling pathways, but additional data would strengthen both claims.

Based on previous experiments (Romero-Carvajal et., 2015) the Notch targets and the reporter are downregulated in response to downregulation of Notch signaling. Therefore, the finding that the reporter and four Notch targets are still robustly expressed in *fgf3* mutants and that hair cell numbers are only increased because of increased progenitor proliferation is evidence that the Notch pathway is not affected in *fgf3* mutants. Unfortunately, the relatively minor gene expression changes between *fgf3* mutants and siblings precluded the detection of signaling pathways. Even in the homeostatic data set, GO term analyses did not reveal signaling pathways active in particular clusters (with few exceptions) likely due to limited coverage as mentioned by the reviewer. None of the Notch pathway genes are significantly changed in the *fgf3* mutants supporting our interpretation that the Notch pathway is not affected by the loss of *fgf3*. However, as we also did not detect changes in Wnt pathway members by scRNASeq, even though in situ hybridization and functional analyses show that Wnt signaling is induced, we feel that citing the unchanged Notch expression in the scRNA seq data is not appropriate in this particular experiment.

9) "fgf3 on the other hand, is slightly downregulated after Notch signaling inhibition with a γ-secretase inhibitor". The data supporting this point seems quite weak. My suggestion is that the authors either remove these data and point from the paper, or else substantiate it better – for example, by treating their neuromast reporter line with γ secretase inhibitors, sorting the cells and measuring fgf3 message by qRT-PCR. It is not an essential point of the paper, so I would be fine with just removing it.

We agree with the reviewer and have deleted the arrow from our model figure and rephrased the interpretation. However, we believe that it is important to show the slight effect of loss of Notch on *fgf3*, as it is reproducible.

10) I recommend Supplementary Figure 10 be included in the main figures, not as a supplementary figure. As mentioned above, I would also be fine with removing the weak dotted line from Notch to fgf3.

We have now transformed Supplementary Figure 10 into main Figure 9 and removed the dotted line from Notch to *fgf3*.

A few points on the Discussion:11) I was not very convinced by the attempts of the authors to bolster the references to stem cells in other systems. For example, the shared expression of tspan1 and its planarian pseudo-homologue is a bit weak. Similarly, in the subsection “Support cells share gene expression profiles with stem cells in other organs”, although GLAST and FABP7 are SVZ markers, they're also expressed in mature astrocytes and some mature mammalian supporting cells. Where the authors state of neuromast supporting cells that "their gene expression profiles share many genes with stem cells in the CNS" – how many is "many"? And what genes are these in addition to the five listed in that paragraph?

We agree with the reviewer and have removed the reference to planarian *tspan1*. However, we believe that our point that a number of genes are shared between lateral line support cells and stem cells in other organs is indicative that these cell populations share similar properties. It does not indicate that the markers themselves are stem cell markers. With respect to the number of shared genes, it would require a more detailed bioinformatic analyses to identify all shared genes, but a significant number of specific support cell cluster markers that we identified reveal associations with stem cells in pub med searches.

12) Subsection “Signals that control multipotency versus differentiation”. Care should be taken in interpretation of negative expression of Notch reporters as being indicative of a lack of Notch signaling, as these reporters can be extremely context dependent, and a negative result with one measurement is potentially problematic. Moreover, work in the mammalian organ of Corti has shown that some supporting cells (inner pillar cells) can have both Notch and FGF signaling active in them, and their conversion to hair cells requires both pathways to be blocked.

We would like to thank the reviewer for this comment. Figure 4—figure supplement 3 shows that other Notch pathway components are expressed in D/V cells. Therefore, more studies are needed to determine if the Notch pathway is indeed absent from the poles as suggested by the Notch reporter. We have therefore deleted this paragraph.

13) Subsection “Fgf3 inhibits Wnt signaling and proliferation possibly viaSost” refers to the role of Wnt signaling in the mammalian organ of Corti. The authors should reference the recent Development paper from the Edge and Dabdoub labs. Also, there are several databases of supporting cell gene expression that the authors could use to look at sclerostin expression in mammals.

We have now added the reference, thank you for the suggestion. We have searched several mammalian data bases and have added the following sentence regarding mammalian Sost expression: ‘As Wnt signaling also induces proliferation in mammalian hair cell progenitors (Chai et al., 2012; Jacques et al., 2012; Jan et al., 2013; Samarajeewa et al., 2018; Shi et al., 2012) a role for Sost paralogs in this system should be tested. According to several data bases, Sost is not, or only lowly expressed in the mammalian ear sensory epithelia, whereas Sostdc is robustly expressed’.

14) Finally, I would recommend deleting the section titled "A role for FGF is (sic) specification?". The points made here are not especially strong and don't contribute significantly to the main messages of the paper.

We agree and deleted this paragraph.

Some additional very minor points:15) In the Abstract, I suggest changing "we implemented a shiny application" to "we implemented a web-based application". It is likely that many people will be ignorant of the Shiny platform and may think this is a typo! Similarly, the reference to the "shiny app" would benefit from a web link.

Corrected.

16) In Figure 1B, I recommend labeling the supporting cells in addition to the hair and mantle cells.

Corrected.

Reviewer #2 (Minor Comments):Two minor comments:- In figure 1K, isl1 is capitalized, should be lower case.

Corrected.

- The act of dissociating zebrafish embryos into single cells is likely to trigger a global injury response. Some discussion of false positive expression of these genes should be discussed.

We have now added a paragraph at the beginning of the Results section discussing these caveats: ‘Dissociating tissues has the caveat that it likely triggers gene expression changes due to loss of adhesion molecules or to a global injury response. We controlled for a global gene expression response by only analyzing genes that are variable between clusters, however cluster-specific gene up- or down regulation can only be controlled for by performing in situ hybridization in intact organs’.